# TTT++: When Does Self-Supervised Test-Time Training Fail or Thrive?

**Yuejiang Liu**     **Parth Kothari**     **Bastien van Delft**

**Baptiste Bellot-Gurlet**     **Taylor Mordan**     **Alexandre Alahi**

École Polytechnique Fédérale de Lausanne (EPFL)

`{firstname.lastname}@epfl.ch`

## Abstract

Test-time training (TTT) through self-supervised learning (SSL) is an emerging paradigm to tackle distributional shifts. Despite encouraging results, it remains unclear when this approach thrives or fails. In this work, we first provide an in-depth look at its limitations and show that TTT can possibly deteriorate, instead of improving, the test-time performance in the presence of severe distribution shifts. To address this issue, we introduce a test-time feature alignment strategy utilizing offline feature summarization and online moment matching, which regularizes adaptation without revisiting training data. We further scale this strategy in the online setting through batch-queue decoupling to enable robust moment estimates even with limited batch size. Given aligned feature distributions, we then shed light on the strong potential of TTT by theoretically analyzing its performance post adaptation. This analysis motivates our use of more informative self-supervision in the form of contrastive learning for visual recognition problems. We empirically demonstrate that our modified version of test-time training, termed *TTT++*, outperforms state-of-the-art methods by significant margins on several benchmarks. Our result indicates that storing and exploiting extra information, in addition to model parameters, can be a promising direction towards robust test-time adaptation. Our code is available at `https://github.com/vita-epfl/ttt-plus-plus`.

## 1 Introduction

Machine learning models often struggle to generalize under distribution shifts. Even a perceptually mild shift between training and test data, *e.g.*, JPEG compression, may cause severe prediction errors [1]. One popular family of methods to address this challenge is to learn an invariant representation across domains by making use of labelled training data and unlabelled test data simultaneously [2–5]. However, revisiting training data at test time can be impractical due to increasing privacy concerns, inflating sizes of datasets as well as many other real-world constraints. This shortcoming prompts a more challenging yet appealing *test-time adaptation* paradigm: given a trained model, how can we adapt it from one domain to another on the fly, without access to training data and human annotations?

One promising approach towards this goal is test-time training (TTT) through self-supervision [6]. The key idea of TTT is simple and straightforward: train the model on two tasks, namely a main task and a self-supervised learning (SSL) task, and update the model based only on the SSL task at test time. This technique implemented with self-supervised rotation prediction has shown encouraging results for improving the robustness of image classifiers under a variety of distributional shifts. Yet, its empirical performance is still inferior to other families of test-time algorithms [7, 8].

35th Conference on Neural Information Processing Systems (NeurIPS 2021).

In this paper, we first take an in-depth look at TTT with emphasis on its limitations. Our analysis starts with a basic question: can TTT always mitigate the effects of distributional shifts? Through an illustrative problem, we show that the TTT framework can lead to surprising failures, deteriorating the test accuracy rather than improving it. This problem is largely attributed to the unconstrained update from the SSL task that interfere with the main task. To address this issue, we introduce a test-time feature alignment strategy by means of offline feature summarization and online moment matching: once training completes, we compute the mean and covariance matrix of training features and store them as part of the model, referred to as *offline feature summarization*; at test time, we encourage the test feature distribution to be close to the training one by matching the moments estimated online with those pre-computed offline, a process referred to as *online moment matching*.

One practical challenge for online feature alignment lies in scaling the strategy to problems with a large number of classes, as obtaining a robust estimate of moments often requires at least a handful of samples per class. To mitigate this issue, we draw inspiration from recent literature [9] and decouple the sample size from the batch size for moment estimates. Specifically, we maintain a large dynamic queue of encoded features and progressively update it in a mini-batch manner. This modification enables effective feature alignment even with limited batch size, greatly improving its viability in the online test-time setting.

Finally, we shed light on the strong potential of TTT through a theoretical analysis of the test accuracy after adaptation. In particular, we derive a lower bound of the test accuracy on the main task and show that it is expected to grow rapidly when the SSL task gets closer to the main task. These findings motivate our integration of contrastive representation learning [9–12], as a strong instance of SSL, into the TTT framework in visual recognition problems.

By combining the three proposed components, we devise an improved version of test-time training, termed *TTT++*. Experimental results show that TTT++ significantly outperforms other recent methods by significant margins on various robustness benchmarks. Our results suggest that exploiting extra information, including both task-specific information in the form of strong self-supervision and domain-specific information in the form of feature summarization, can be a promising direction to enhance the effectiveness of test-time adaptation.

## 2 Background

### 2.1 Related Work

**Test-time Adaptation.**   Adapting machine learning models based on test samples has garnered growing interests in both generative problems such as super-resolution [13], image synthesis [14] and image manipulation [15], and discriminative problems like image classification [16]. Our work is focused on the latter one in the presence of distributional shifts. Several recent works [16, 17] have shown the advantage of adapting the learned classifier to new test domains in the unsupervised manner, without revisiting the source data. One simplest form is to replace the batch-norm statistics estimated on the training set with those on test examples [17]. Another line of work proposed to adapt the model parameters by exploiting the predicted labels on test examples, such as entropy minimization [8] and pseudo-labeling [7]. While these methods yield promising results on some benchmarks, they are inherently restricted to classification problems and often vulnerable under large distribution shifts [18].

More closely related to ours, [6] proposed test-time training through self-supervised learning, *e.g.*, predicting the type of image rotations. This approach does not involve any assumptions about the output for the main task and is therefore more generic. It has been successfully applied to a variety of problems, such as instance tracking [19] and reinforcement learning [20]. Nevertheless, it was shown empirically inferior to other test-time algorithms [8]. Our work provides an in-depth analysis of its limitations, introduces simple yet effective remedies, and consolidates more theoretical grounds.

**Feature Alignment.**   Aligning the distributions of training and test samples in the feature space is commonly used for domain adaptation. Previous feature alignment methods fall into two main categories: minimizing a divergence measure, such as MMD [21], Coral [22] and CMD [23], or encouraging the domain confusion through adversarial training [5, 24]. However, most of these methods rely on the co-existence of source and target data, and thus cannot be readily applied to the test-time setting where source data is not available. Our work revisits the critical role of feature

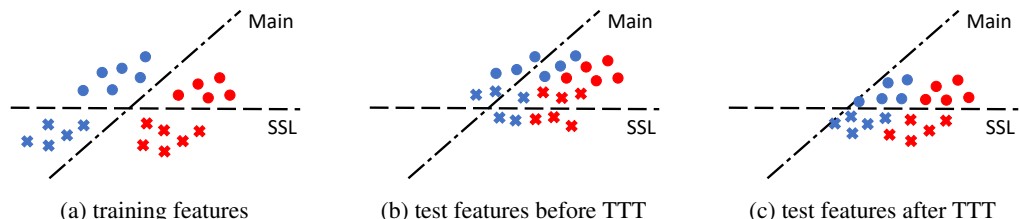

(a) training features       (b) test features before TTT       (c) test features after TTT

Figure 1: Illustration of a failure case where TTT hurts robustness under distributional shifts. (a) The predictive model reaches high accuracy on both the main classification task, *i.e.*, separating red and blue data points, and the auxiliary self-supervised task, *i.e.*, separating circles and crosses, in the training domain. (b) Given a large distributional shift, test samples are encoded into a new subspace, resulting in limited accuracy on both tasks. (c) Without any constraints on the feature distribution, TTT may result in an updated encoder severely overfitting to the SSL task, and consequently deteriorate the accuracy on the main task, as opposed to improving it.

alignment for test-time training and proposes a simple and practical strategy that enables online feature alignment, even with a limited batch size.

**Self-Supervised Learning.** Self-supervised learning is a powerful paradigm to learn rich representations from unlabeled samples. Stunning progress has been made in recent years by designing informative self-supervised tasks [11, 12, 25–28] and stabilizing the training process [9, 29]. Prior works are mainly focused on unsupervised pre-training, whereas our work looks into the importance of incorporating strong self-supervised learning methods for test-time adaptation.

## 2.2 Preliminary: Test-Time Training

Test-time training (TTT) [6] is a general framework for adapting neural network models to a new test distribution based on unlabeled samples. Different from the conventional approach that trains the model only on the task of interest, TTT considers two tasks: a main task and an auxiliary SSL task. The model is trained on both tasks simultaneously with a multi-task architecture composed of one shared encoder $g$ and two separate heads $\pi_m$ and $\pi_s$ respectively. Given a labeled training dataset $D = \{(x_i, y_i)\}_{i \in \{1,...,N\}}$, the model is trained to minimize two losses jointly:

$$\mathcal{L}_{train}(D; g, \pi_m, \pi_s) = \frac{1}{N} \sum_{i=1}^{N} \mathcal{L}_m(x_i, y_i; g, \pi_m) + \lambda \mathcal{L}_s(x_i; g, \pi_s), \tag{1}$$

where $\lambda$ is a hyper-parameter to balance the two tasks.

In the presence of distributional shifts, the learned model often struggles to directly generalize to a new test set $D' = \{x_i'\}_{i \in \{1,...,N'\}}$. The core idea of TTT is to fine-tune the encoder $g$ based on the self-supervised task with the test examples,

$$\mathcal{L}_{TTT}(D'; g') = \frac{1}{N'} \sum_{i=1}^{N'} \mathcal{L}_s(x_i'; g'), \tag{2}$$

with the hope that the updated model $\pi_m \circ g'$ yields improved results on the main task.

TTT instantiated with a self-supervised rotation prediction task has been demonstrated effective for improving the robustness of image classifiers under common distributional shifts [6]. However, it was shown inferior to other families of test-time adaptation methods [7, 8]. We will next look into its strengths and limitations, and propose an improved version for better adaptation performance.

## 3 When Does Test-Time Training Fail?

In this section, we first throw light on a caveat of test-time training under large distributional shifts, and subsequently introduce practical solutions tailored for the test-time setting.

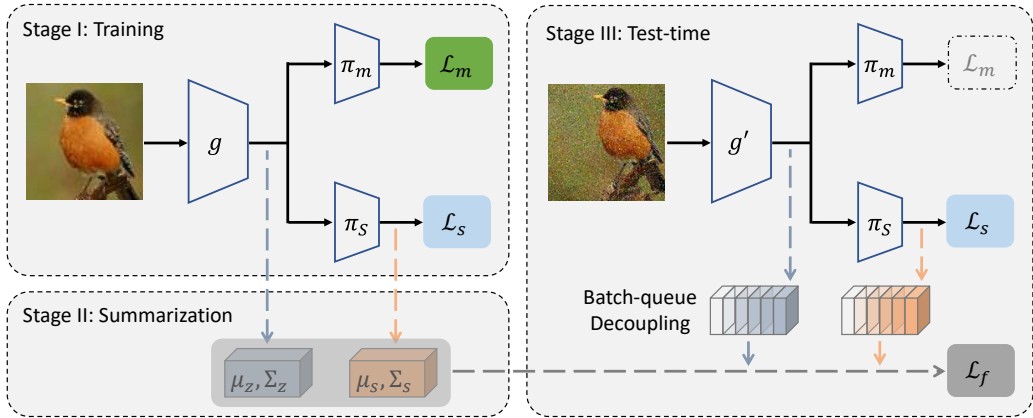

Figure 2: Our modified version of test-time training (TTT++). Our method consists of three stages: model training, offline feature summarization, and online test-time adaptation. (i) During training, the model is optimized for the main task and an auxiliary contrastive self-supervised task jointly. (ii) Once training completes, we summarize the feature distributions after the encoder and the self-supervised head in the form of first and second-order moments. (iii) At test time, we adapt the encoder through online feature alignment (Sec. 3.2) and self-supervised learning (Sec. 4.2). In case of limited batch size, we maintain a large dynamic queue of feature vectors for robust moment estimates (Sec. 3.3).

### 3.1 Illustrative Example of Failures

One implicit assumption behind TTT is that the encoder update based on SSL can counter the effect of the underlying distributional shift on the main task. This assumption, however, is likely broken under large shifts and results in unexpected adaptation failures. To illustrate this limitation, we introduce a simple toy problem in Figure 1, where the main task and the SSL task are defined as classifying colors and symbols of encoded features in the 2-dimensional latent space. We consider an ideal scenario where the two tasks are highly correlated such that a well-trained model can attain high accuracies on both of them in the training domain. Nevertheless, in the presence of a significant distributional shift, the model may still suffer from substantial prediction errors in the test domain.

In this scenario, while TTT may restore the discriminative power of the learned representation for the main task to a certain degree, the unconstrained self-supervised adaptation may yield severe overfitting to the auxiliary SSL task. As a consequence, the performance on the main task can even deteriorate as opposed to improving. This phenomenon is not restricted to our illustration and also occurs in practice, as shown in Section 5.1.

### 3.2 Online Feature Alignment

As illustrated in the toy example above, simply applying self-supervised adaptation at test time can lead to arbitrarily poor results. To address this issue, we introduce an online feature alignment strategy to ensure robust adaptation at test time. The core idea of our strategy is to impose a constraint over the feature space during TTT such that the feature distribution of test examples remains close to that in the training domain. While some feature alignment techniques such as MMD [2] and adversarial training [24] have been widely used for domain adaptation, they often rely on sampling from training and test domains concurrently, which is impractical in the test-time setting. We, therefore, turn to classical divergence measures that can be estimated independently for each distribution. More specifically, we use the square distance of the first and second moments between two feature distributions, inspired by DDC [30] and Coral [22], to approximate the domain discrepancy. This design choice allows us to summarize the distribution of training features in a compact format and store it as part of the model, eliminating the need to revisit the training data during test-time adaptation.

Concretely, once training completes, we perform an offline feature summarization step that characterizes the distribution of feature vectors in the training domain $Z = \{z_1^T, \ldots, z_N^T\}$ through the empirical mean $\mu_z = \frac{1}{N} \sum_i^N z_i$ and covariance matrix $\Sigma_Z = \frac{1}{N-1}(Z^T Z - (I^T Z)^T (I^T Z))$. The former is essentially equivalent to the channel-wise batch normalization statistics while the latter is more informative yet light-weight for computation and storage. At test time, we regularize the

self-supervised adaptation by minimizing the distance between the feature statistics estimated from a mini-batch of test samples, *i.e.*, $\mu_z'$ and $\Sigma_z'$, and the pre-stored quantities about the training domain:

$$\mathcal{L}_{f,z} = \|\mu_z - \mu_z'\|_2^2 + \|\Sigma_z - \Sigma_z'\|_F^2 \,, \tag{3}$$

where $\|\cdot\|_2$ is the Euclidean norm and $\|\cdot\|_F$ is the Frobenius norm.

The basic form of online moment matching can be limiting in that the low-order statistics may be insufficient to fully capture complex distributions in high dimensions, *e.g.*, 2048 for the standard ResNet-50. To alleviate this issue, we align the feature distributions at both the output of the encoder and the output of the self-supervised head, which are of lower dimensions, *e.g.*, 128 in the case of contrastive learning described in Sec. 4.2. Our final objective at test time is a weighted combination of the self-supervised loss $\mathcal{L}_s$, the feature alignment loss at the encoder $\mathcal{L}_{f,z}$ and that at the self-supervised head $\mathcal{L}_{f,s}$,

$$\mathcal{L}_{TTT++} = \mathcal{L}_s + \lambda_z \mathcal{L}_{f,z} + \lambda_s \mathcal{L}_{f,s}, \tag{4}$$

where $\lambda_z$ and $\lambda_s$ are hyper-parameters controlling the emphasis on each term.

### 3.3 Online Dynamic Queue

One practical challenge for online feature alignment lies in scaling the strategy to problems having large numbers of classes. Intuitively, a good estimate of moments of the entire distribution needs at least a handful of samples per class. As a consequence, the demand for sample size grows linearly with the number of classes, for instance, over $\sim$1000 samples are required in the case of CIFAR-100. However, the computational resources during deployment are often limited to accommodate such a large batch size in the test-time setting.

To overcome this challenge, we draw inspiration from recent literature [9] and maintain a dynamic queue of encoded features to decouple the batch size from the sample size for moment estimates. More specifically, we construct a dynamic queue that contains a few batches of feature vectors encoded at test time. We progressively update the queue by appending the latest mini-batch and popping out the oldest one, as illustrated in Figure 2. This batch-queue decoupling allows us to collect a large and consistent pool of samples for online moment matching, even with a very limited batch size.

By integrating the online moment matching and batch-queue decoupling, our test-time algorithm takes into account both the discriminative power and the marginal distribution of the updated representations, enabling more robust adaptation under various settings. It is worth noting that the current moment matching strategy is just a particular instance of the online feature alignment scheme. It can be naturally extended to incorporate higher-order statistics [23, 31] to bring further performance gain at the cost of larger space and computational complexities.

## 4  When Does Test-Time Training Thrive?

Given properly aligned feature distribution, we next look into the potential of test-time training given strong SSL tasks. We first derive a lower bound of the test accuracy in general scenarios and then analyze it in a specific setup where the performance on the main task can be directly estimated. These analyses motivate our use of more informative self-supervised learning for test-time training.

### 4.1 Theoretical Results

We consider a training set comprised of samples drawn from the joint distribution $\mathbb{P}_{X,Y_m,Y_s}$, where $X, Y_m$ and $Y_s$ are random variables corresponding to the training samples, the main task labels and the self-supervised labels respectively. Similarly, the test set consists of samples drawn from the joint distribution $\mathbb{P}_{X',Y_m',Y_s'}$. In the presence of distribution shift, $\mathbb{P}_{X,Y_m,Y_s}$ and $\mathbb{P}_{X',Y_m',Y_s'}$ are not identical. However, we make the following assumption about label distribution in our analysis.

**Assumption 1.** *The training and test labels are equal in distribution,* $Y_m \overset{d}{=} Y_m'$, $Y_s \overset{d}{=} Y_s'$ *and* $(Y_m, Y_s) \overset{d}{=} (Y_m', Y_s')$.

In addition, we restrict our analysis to the scenarios where both two tasks can be solved perfectly during training.

**Assumption 2.** *There exist an encoder g and classifiers $\pi_m$ and $\pi_s$ such that $\mathbb{P}(\pi_m(g(X)) = Y_m) = 1$ and $\mathbb{P}(\pi_s(g(X)) = Y_s) = 1$.*

During TTT, the shared encoder $g$ is updated to $g'$ such that the self-supervised head $\pi_s$ fits the test data. Given our proposed online feature alignment in Equation 4, we assume the encoded features in the training and test domains, *i.e.*, $Z$ and $Z'$, have the following property.

**Assumption 3.** *The marginal feature distribution and the conditional feature distributions at test time are aligned with their counterparts during training, that is, $Z \overset{d}{=} Z'$ and $(Z \mid Y_s = k) \overset{d}{=} (Z' \mid Y'_s = k)$ for all classes $k$ in the SSL task.*

Under these assumptions, we consider the outcome of test-time training in the worst-case scenario.

**Theorem 1.** *The prediction accuracy on the main task is lower bounded:*

$$\mathbb{P}(\pi_m(Z') = Y'_m) \geq \sum_{y_s} \mathbb{P}(Y_s = y_s) \max \left\{ 0, 2 \left( \max_{y_m} \mathbb{P}(Y_m = y_m \mid Y_s = y_s) - 0.5 \right) \right\}. \quad (5)$$

*Proof.* Please refer to Section A.1 in the supplementary material. □

Theorem 1 highlights the importance of the relation between the main task and the SSL task for the test accuracy after adaptation. More specifically, the SSL task needs to be informative with respect to the main task to guarantee the performance, *i.e.*, knowing the SSL class $y_s$ makes a main class $y_m$ highly probable, or equivalently $\mathbb{P}(Y_m = y_m \mid Y_s = y_s)$ is large, leading to a greater lower bound.

To further understand the impact of the task relation on test-time training, we next consider a particular setting, where the encoded features fully overfit to the SSL task (*e.g.*, lengthy test-time training) and become independent of the main task label given the SSL label. Under this condition, the prediction accuracy on the main task can be directly estimated as follows.

**Theorem 2.** *If $Z' \perp\!\!\!\perp Y'_m \mid Y'_s$, then the prediction accuracy on the main task is given by*

$$\mathbb{P}(\pi_m(Z') = Y'_m) = \sum_{y_s} \left[ \mathbb{P}(Y_s = y_s) \sum_{y_m} \mathbb{P}(Y_m = y_m \mid Y_s = y_s)^2 \right]. \quad (6)$$

*Proof.* Please refer to Section A.2 in the supplementary material. □

Intuitively, if the encoded features do not contain more information than the SSL labels about the main task, the accuracy of the main classifier $\pi_m$ only depends on the property of the SSL task. In particular, the square term on the right-hand side of Equation 6 emphasizes the paramount importance of designing a closely related SSL task. When the two tasks diverge, the test accuracy drops quadratically fast, leading to ineffective adaptation.

## 4.2 Test-Time Training through Contrastive Learning

Our theoretical analysis above reveals the importance of incorporating an SSL task highly correlated with the main task for test-time training. One practical way to quantify the relation between two tasks is to measure the transferability of the representation learned from one task to another [32]. Given the remarkable results of contrastive methods for visual representation pre-training [9, 12, 33, 34], we hypothesize that they would also be suitable choices for test-time training. We thus replace the rotation prediction task with SimCLR [12] in the context of visual recognition. Given a mini-batch of $B$ images, we augment each image to two views. We consider the two augmented views from the same original instance as a positive pair and treat the other pairs as negative ones. The feature vector $z_i = g(x_i)$ of each image $x_i$ is projected to a lower-dimensional space $h_i = \pi_s(z_i)$ through our self-supervised head. The projected hidden embeddings from a positive pair $<h_i, h_j>$ are encouraged to be closer than those from the negative ones through the following loss,

$$\mathcal{L}_s = -\log \frac{\exp(\text{sim}(h_i, h_j)/\tau)}{\sum_{k=1}^{2B} \mathbb{1}_{k \neq i} \exp(\text{sim}(h_i, h_k)/\tau)}, \quad (7)$$

where $\tau$ is a temperature scaling parameter. The distance between projected embeddings is measured by cosine similarity $\text{sim}(u, v) = u^T v / (\|u\| \|v\|)$.

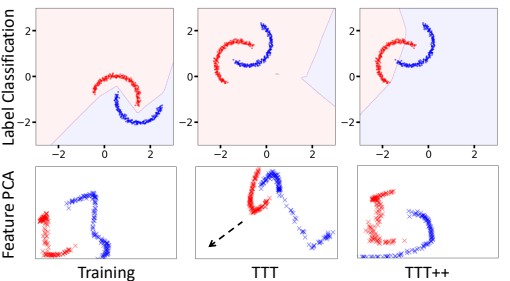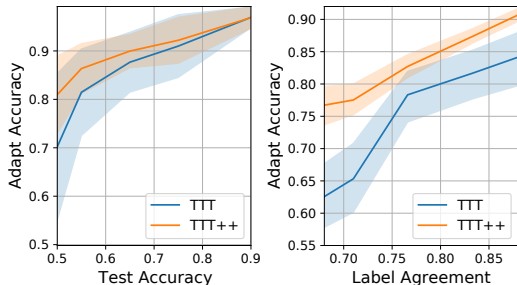

Figure 3: Qualitative comparison of TTT [6] and our TTT++ on the inter-twinning moons problem in the presence of large translation and rotation shifts. The vanilla TTT drives the decision boundary further away from a desired one due to a severe feature misalignment, marked by the dashed arrow in the feature PCA visualization. In comparison, our method adapts the decision boundary to the test domain more effectively.

Figure 4: Quantitative comparison of TTT [6] and our TTT++ on the inter-twinning moons problem under 150 different setups. Standard deviations are visualized in shaded regions. Our method (left) is particularly advantageous under large shifts, *i.e.*, low test accuracy before adaptation, and (right) greatly benefits from a higher correlation, *i.e.*, larger label agreement, between the main and SSL tasks.

## 5 Experiments

We empirically validate our proposed method in four scenarios: synthetic toy problem, common image corruptions, natural domain shifts, and sim-to-real transfer.

We consider the following baselines throughout our experiments:

- **Test**: the model in the training domain is directly evaluated on the test data without any adaptation;
- Test-time normalization (**BN**) [35] updates the batch normalization statistics of the trained network according to the test data;
- Test-time entropy minimization (**TENT**) [8] updates the batch normalization statistics of the trained network by minimizing the entropy of the model predictions on the test data;
- Source Hypothesis Transfer (**SHOT**) [36] freezes the classifier module and updates only the feature extraction module by exploiting the concepts of information maximization and self-supervised pseudo-labeling during testing;
- Test-time training (**TTT-R**) [6] trains the network jointly on the main task and a rotation-based SSL task in the source domain; during test, TTT-R continues to train on the rotation-based task in the target domain.

We also evaluate the following ablated versions of our method:

- Test-time feature alignment (**TFA**) aligns the first-order and second-order statistics of the source and target distributions during testing (Section 3.2 and 3.3);
- Test-time contrastive learning (**TTT-C**) trains the network jointly on the main task and a contrastive learning (SSL) task in the source domain; during test, TTT-C continues to update the encoder based on contrastive learning in the target domain (Section 4.2).

The full version of our method, improved test-time training (**TTT++**), combines TFA and TTT-C.

### 5.1 Synthetic Toy Problem

We first evaluate our method on the inter-twinning moons problem [4, 37], where the main task is to predict the moon class of a given data point and the SSL task is to predict on which side of the hyperplane (*i.e.*, linear separator between the two moons) the data point lies on. The relation (label agreement) between the two tasks depends on the separation distance between the two moons. To solve both tasks simultaneously, we build a small neural network that consists of a 2-layer MLP as the shared encoder and two 2-layer MLPs as separate task heads. Each hidden layer contains 8 neurons. The learned model attains over 99% accuracy on both the main and SSL tasks in the training domain. We simulate a variety of distributional shifts through translation and rotation of all data points.

Figure 3 shows the decision boundaries and encoded features from the vanilla TTT and our proposed TTT++ in a particular test case of large distributional shift. TTT not only fails to improve the classification accuracy but even pushes the decision boundary further away from a desired one due to

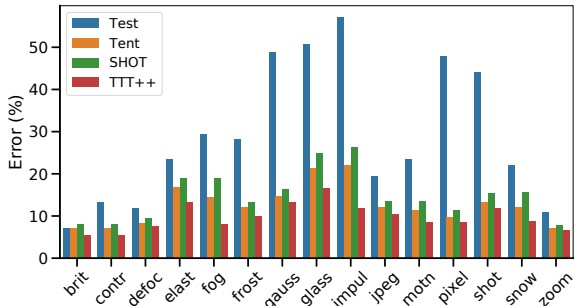

Figure 5: Classification error (%) on CIFAR10-C [1].

Table 1: Average classification error (%) on CIFAR10-C/100-C [1] and CIFAR10.1 [40]

| Method | C10-C | C100-C | C10.1 |
|---|---|---|---|
| Test | 29.1 | 61.2 | 12.1 |
| BN [41] | 15.7 | 43.3 | 14.1 |
| TTT-R [6] | 14.3 | 40.4 | 11.0 |
| SHOT [36] | 14.7 | 38.1 | 11.1 |
| TENT [8] | 12.6 | 36.3 | 13.4 |
| TFA (Ours) | 11.9 | 35.8 | 12.1 |
| TTT-C (Ours) | 10.7 | 36.9 | 9.7 |
| TTT++ (Ours) | **9.8** | **34.1** | **9.5** |

the severe feature distribution mismatch, as evidenced in the PCA visualization. In comparison, our TTT++ yields substantial performance gain, boosting the test accuracy from 50% to 93%, thanks to the reduced feature distributional shift enforced by the proposed online moment matching.

Figure 4 summarizes the quantitative results of our methods as well as the vanilla counterpart under 150 different simulated setups. As shown on the left graph, when the *domain shift* only causes mild test errors on the main task, both the original TTT and our TTT++ yield strong adaptation results. However, the effectiveness of TTT deteriorates quickly along with the growth of domain shift, as reflected on the lower test accuracy. In comparison, our proposed TTT++ demonstrates clear advantages under large shifts, *e.g.*, when the test accuracy before adaptation is around 0.5. We further examine the impact of the relation between the main and SSL tasks on the adaptation performance by varying the separation distance between the two moons. The simulation results confirm the high potential of test-time training given improved SSL tasks, as analyzed in Section 4.1.

## 5.2 Common Image Corruption

We further assess the robustness of our method against common image corruptions. Following the evaluation protocol of previous work [8], we train ResNet-50 [38] on CIFAR10/CIFAR100 [39] and test it on the CIFAR10-C/CIFAR100-C [1] datasets, which contain 15 types of *algorithmically generated* corruptions, such as noise, blur and snow effects. We use a batch size of 256 for test-time adaptation. In addition, we use a dynamic queue containing 16 batches of feature vectors for online feature alignment on CIFAR100-C.

Figure 5 shows the quantitative results under each type of image corruption. The average results on CIFAR10-C and CIFAR100-C are reported in Table 1. Our TTT++ clearly outperforms the prior state-of-the-art test-time methods on CIFAR10-C and CIFAR100-C. In particular, incorporating a strong self-supervision task (TTT-C) already suffices to perform on par or better than TENT. Adding test-time feature alignment (TFA) on top of that yields an additional $\sim 8\%$ relative reduction in terms of the test error.

## 5.3 Natural Domain Shift

We next demonstrate the efficacy of our method TTT++ to tackle *natural* distribution shifts. We again use the pre-trained ResNet-50 and test it on CIFAR10.1 [40], a recently collected test set subject to natural distributional shift. Despite its high perceptual similarity with the CIFAR10 dataset, the CIFAR10.1 typically leads to a drop of accuracy (4% to 10%) for a wide range of deep models [40].

Table 1 summarizes the results of various test-time algorithms on CIFAR10.1. Previous batch-norm-based methods perform poorly and even degrade model accuracy. This phenomenon is tied to their implicit assumption that different samples and spatial locations are shifted in a similar manner [18], which is true for algorithmically generated image corruptions but does not hold on CIFAR-10.1. In contrast, TTT++ is more generic and yields stronger performance under the natural distribution shift.

## 5.4 Sim-to-Real Transfer

We finally validate the effectiveness of our method on the VisDA-C dataset [42], a challenging large-scale benchmark of synthetic-to-real object classification. As shown in Table 2, prior methods

Table 2: Classification error (%) on the large-scale VisDA-C dataset [42].

| Method | plane | bcycl | bus | car | horse | knife | mcycl | person | plant | sktbrd | train | truck | Per-class |
|---|---|---|---|---|---|---|---|---|---|---|---|---|---|
| Test | 56.52 | 88.71 | 62.77 | **30.56** | 81.88 | 99.03 | 17.53 | 95.85 | 51.66 | 77.86 | 20.44 | 99.51 | 58.72 |
| BN [41] | 44.38 | 56.98 | 33.24 | 55.28 | 37.45 | 66.60 | 16.55 | 59.02 | 43.55 | 60.72 | 31.07 | 82.98 | 48.12 |
| TENT [8] | 13.43 | 77.98 | **20.17** | 48.15 | 21.72 | 82.45 | 12.37 | 35.78 | 21.06 | 76.41 | 34.11 | 98.93 | 42.73 |
| SHOT [36] | 5.73 | **13.64** | 23.33 | 42.69 | 7.93 | 86.99 | 19.17 | **19.97** | 11.63 | 11.09 | 15.06 | **43.26** | 25.04 |
| TFA (Ours) | 28.25 | 32.03 | 33.67 | 64.77 | 20.49 | **56.63** | 22.52 | 36.30 | 24.84 | 35.20 | 25.31 | 64.24 | 39.58 |
| TTT-C (Ours) | 5.46 | 32.23 | 25.42 | 37.03 | 7.84 | 85.20 | 9.14 | 23.80 | 11.72 | 11.00 | 7.74 | 56.87 | 25.72 |
| TTT++ (Ours) | **4.13** | 26.20 | 21.60 | 31.70 | **7.43** | 83.30 | **7.83** | 21.10 | **7.03** | **7.73** | **6.91** | 51.40 | **22.46** |

Table 3: Classification error (%) results from online feature alignment with or without a dynamic queue of feature vectors on the CIFAR100-C under level-5 fog corruption. Sample size = Batch size × # Batches. Given a fixed batch size, enlarging the queue size leads to similar results as having a larger batch size.

| | w/o queue | | | w/ queue | | | |
|---|---|---|---|---|---|---|---|
| Sample Size | 64 | 128 | 256 | $64 \times 2$ | $64 \times 4$ | $64 \times 8$ | $64 \times 16$ |
| Test Error | 40.31 | 38.67 | 37.01 | 39.84 | 37.37 | 36.18 | 36.02 |

that are fairly competitive under image corruptions, such as BN [41] and TENT [8], are not effective on VisDA-C. We conjecture that this is attributed to their strong restrictions over the adaptable parameters at test time. In contrast, our proposed method is more flexible, allows the model to update the entire encoder, and thus achieves compelling results on VisDA-C. Furthermore, the test-time feature alignment plays a crucial role in this synthetic-to-real domain adaptation problem, providing $\sim 13\%$ performance boost on top of the TTT-C.

## 5.5    Effect of Batch-Queue Decoupling

To verify the effects of batch-queue decoupling, we compare the results of online feature alignment with different sample sizes. Table 3 summarizes the test errors on CIFAR100 under the level-5 fog corruption. As expected, larger sample sizes generally lead to lower classification errors. Interestingly, while the performance of using a dynamic queue is slightly worse than its counterpart of the same sample size from a single large batch, enlarging the queue size always yields stronger results. For instance, given a small batch size of $64$, using a dynamic queue maintaining $512$ or $1024$ feature vectors from 8 or 16 consecutive batches respectively is more advantageous than the vanilla moment matching based on a batch size of $256$ samples. This result corroborates the benefit of integrating a dynamic queue into our proposed feature alignment framework, for enhancing the scalability in the online setting.

## 5.6    Design Choice for Moment Matching

As discussed in Section 3, our proposed online feature alignment can be instantiated with different orders of moments and applied at various layers. To understand the effects of the detailed design choices, we empirically compare our proposed version against several ablated variants in Table 4. Irrespective of the layer choice, our proposed online feature alignment consistently results in reduced classification error. Nevertheless, moment matching applied to the self-supervised head alone leads to lower test error compared to applying it to the feature extractor output in 11 out of 15 types of corruption. We conjecture that the strong performance of the former one is attributed to the lower dimensionality of the feature vector, which allows for a more accurate estimate of feature statistics. The best result comes from the online feature alignment at the outputs of both the encoder and the projection head, which validates our design choice in Section 3.2.

We further validate the choice of divergence measure through an ablation study. The online feature alignment using the second-order moment (covariance) leads to clearly better results than the one using the first-order moment (mean). It is also evident that the online feature alignment is most effective when both the mean and the covariance are taken into account.

Table 4: Classification error (%) on CIFAR10-C [1] with different versions of online feature alignment. Taking into account both the first and second-order moments at two different layers is better than the other counterparts in terms of the robustness against most types of image corruptions.

| TFA | brit | contr | defoc | elast | fog | frost | gauss | glass | impul | jpeg | motn | pixel | shot | snow | zoom |
|---|---|---|---|---|---|---|---|---|---|---|---|---|---|---|---|
| w/o $\mathcal{L}_{f,s}$ | 7.96 | 7.57 | 9.4 | 17.24 | 14.38 | 12.54 | 14.62 | 21.05 | 21.4 | 12.68 | 11.92 | 10.7 | 13.7 | 12.74 | 7.32 |
| w/o $\mathcal{L}_{f,z}$ | 7.85 | 7.84 | 9.18 | 16.51 | 14.33 | 11.99 | 13.79 | 20.08 | 20.17 | 12.42 | 12.02 | 10.5 | 12.78 | 13.28 | 7.44 |
| w/o $\Sigma$ | 7.49 | 7.56 | 9.62 | 18.62 | 19.22 | 12.72 | 16.02 | 25.07 | 25.17 | 13.43 | 13.63 | 11.22 | 15.04 | 15.11 | 7.77 |
| w/o $\mu$ | **7.43** | **7.37** | 8.90 | 15.92 | 12.98 | 11.57 | 13.46 | 19.27 | **18.95** | 11.87 | 11.11 | 9.97 | 12.81 | 11.76 | 7.04 |
| Full | 7.44 | 7.40 | **8.89** | **15.73** | **12.82** | **11.49** | **12.94** | **18.46** | 19.13 | **11.66** | **10.77** | **9.93** | **12.67** | **11.73** | **7.03** |

## 6 Conclusion and Discussions

In this work, we conduct an in-depth analysis of the limitations and potential of self-supervised test-time training. We draw attention to the risk of feature distribution mismatch, which is critical but largely overlooked in recent test-time algorithms. We shed light on the strong potential of this approach by analyzing the growth of test accuracy given improved SSL tasks. These analyses inspire three proposed modifications, namely online feature alignment, batch-queue decoupling and contrastive test-time training, which yield state-of-the-art results on multiple robustness benchmarks. Our results suggest the advantages of bringing additional task-specific and domain-specific information in a compact format for test-time adaptation. We hope these findings will motivate researchers and practitioners to *rethink what should be stored*, in addition to weight parameters, for the robust deployment of machine learning models.

**Limitations.** In this work, we restrain feature summarization to first and second-order moments. Yet, the low-order statistics may be insufficient to characterize the complex distribution of high-dimensional features. Developing more advanced summarization methods tailored for test-time adaptation is an interesting avenue for future work. In addition, there may exist a considerable gap between our theoretical analysis and the empirical results, when the stated assumptions do not hold. For instance, neither the classification heads nor the feature alignment is perfect in practice. More theoretical guarantees can be valuable for the practical use of test-time adaptation in safety-critical scenarios.

**Open Questions.** In our experiments, we only consider the standard ResNet-50 as the backbone architecture and share the whole feature extractor between the main task and self-supervised task. Yet, recent literature [30, 43] has shown that different layers capture different levels of semantic granularity. The impact of architectural design on test-time training remains an open question. Furthermore, while we empirically compare our proposed method against other families of test-time adaptation algorithms, these techniques exploit different supervisory signals extracted from unlabeled data, which can be complementary to each other. Blending these techniques into a unified framework is another interesting direction to explore in the future.

**Societal Impact.** Our work aims at expanding the current horizon of machine learning algorithms for test-time adaptation. For applications where humans' lives are at risk, such as autonomous driving, trust, safety, robustness are all mandatory keywords. The field has made amazing progress when the training and testing environments are highly similar. What happens when a machine gets deployed in a new environment? We, humans, have an innate capability for handling such shifts. We believe that machines should have the same capability as humans. Indeed, there is a long way to go. Nevertheless, we hope that our work will foster more research in analyzing and devising algorithms for robust and effective adaptation at test time.

## Acknowledgements

This work was supported by the Swiss National Science Foundation under the Grant 2OOO21-L92326, Honda R&D Co. Ltd, EPFL Open Science fund and Valeo. We thank Sudeep Salgia, Tao Lin, Lingjun Meng, Yifan Sun for helpful inputs to our early drafts and anonymous reviewers for valuable comments.

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
