# A  Proofs of Theorems

In this section, we prove the theoretical results in the Section 4.1.

## A.1  Proof of Theorem 1

**Lemma 1.** *Let $(\Omega, \mathcal{F}, \mathbb{P})$ be a probability space and let $(A_i)_{i \in \{1...n\}}$ be a partition of $\Omega$. Let $C$ be the set of partitions of $\Omega$ whose elements have the same probabilities as $(A_i)_{i \in \{1...n\}}$, that is:*

$$C = \{(U_i)_{i \in \{1...n\}} \, / \, \bigcup_i U_i = \Omega; \quad \forall (i,j), i \neq j, U_i \cap U_j = \varnothing; \quad \forall i, \mathbb{P}(U_i) = \mathbb{P}(A_i)\}. \quad (8)$$

*If $n = 2$ or $\mathbb{P}(A_1) \geq 1/2$ then:*

$$\min_{(B_i)_{i \in \{1...n\}} \in C} \sum_i \mathbb{P}(B_i \cap A_i) \geq 2 \times P(A_1) - 1. \quad (9)$$

*Proof.* If $n > 2$ and $\mathbb{P}(A_1) \geq 1/2$, then we can write $A_1' = A_1$ and $A_2' = (\bigcup_{i=2...,n} A_i)$ and reason similarly as in the case where $n = 2$ with $(A_1', A_2')$ and $(B_1', B_2')$.

In the case n = 2, we have:

$$\begin{aligned}
\mathbb{P}(A_1 \cap B_1) + \mathbb{P}(A_2 \cap B_2) &= 1 - \mathbb{P}(A_1 \cap B_2) - \mathbb{P}(A_2 \cap B_1) \\
&\geq 1 - \mathbb{P}(B_2) - \mathbb{P}(A_2) \\
&\geq 1 - 2 \times (1 - P(A_1)) \\
&\geq 2 \times \mathbb{P}(A_1) - 1
\end{aligned}$$

The intuition is that no matter how the partitions are built, if $P(A_1) > 1/2$ and $P(B_1) > 1/2$, there is necessarily an overlap between the two subsets such that $A_1 \cap B_1 \neq \varnothing$. $\qquad \square$

**Theorem 1.** *If we assume that:*

- $Z \overset{d}{=} Z'$;

- $(Z \mid Y_s = k) \overset{d}{=} (Z' \mid Y_s' = k), \forall k \in \{1, ..., K\}$;

*then the accuracy of the main task classifier is lower-bounded:*

$$\mathbb{P}(\pi_m(Z') = Y_m') \geq \sum_{y_s} \mathbb{P}(Y_s = y_s) \max \left\{ 0, 2 \left( \max_{y_m} \mathbb{P}(Y_m = y_m \mid Y_s = y_s) - \frac{1}{2} \right) \right\}. \quad (10)$$

*Proof.* Using that $\bigcup_{y_s} \{Y_s' = y_s\} = \Omega$, we obtain using the law of total probability that:

$$\overset{\overbrace{\text{Accuracy}}}{\mathbb{P}(\pi_m(Z') = Y_m')} = \sum_{y_s} \mathbb{P}(\pi_m(Z') = Y_m' \mid Y_s' = y_s) \mathbb{P}(Y_s' = y_s)$$

$$= \sum_{y_s} \mathbb{P}(Y_s' = y_s) \sum_{y_m} \mathbb{P}(\{\pi_m(Z') = y_m\} \cap \{Y_m' = y_m\} \mid Y_s' = y_s). \quad (11)$$

Let us introduce $A_{y_m}^{(y_s)} = \{Y_m' = y_m \mid Y_s' = y_s\}$ and $B_{y_m}^{(y_s)} = \{\pi_m(Z') = y_m \mid Y_s' = y_s\}$. It is important to note that $\mathbb{P}(A_{y_m}^{(y_s)}) = \mathbb{P}(B_{y_m}^{(y_s)})$. Indeed, we know that:

$$\mathbb{P}(B_{y_m}^{(y_s)}) = \mathbb{P}(\pi_m(Z') = y_m \mid Y_s' = y_s). \quad (12)$$

Moreover, we assume conditional distribution to be aligned, and $\pi_m$ not to be retrained, as a result equation (12) can be written as:

$$
\begin{aligned}
\mathbb{P}(B_{y_m}^{(y_s)}) &= \mathbb{P}(\pi_m(Z) = y_m \mid Y_s = y_s) \\
&= \mathbb{P}(Y_m = y_m \mid Y_s = y_s) \\
&= \mathbb{P}(Y_m' = y_m' \mid Y_s' = y_s') \\
&= \mathbb{P}(A_{y_m}^{(y_s)}).
\end{aligned}
\tag{13}
$$

Then, we can rewrite equation (11), namely the accuracy, as:

$$
\overbrace{\mathbb{P}(\pi_m(Z') = Y_m')}^{\text{Accuracy}} = \sum_{y_s} \mathbb{P}(Y_s' = y_s) \sum_{y_m} \mathbb{P}(A_{y_m}^{(y_s)} \cap B_{y_m}^{(y_s)}).
\tag{14}
$$

Finally, without loss of generality, we can assume that the indexes of $(A_{y_m}^{(y_s)})_{y_m}$ and $(B_{y_m}^{(y_s)})_{y_m}$ are ordered such that:

1. $\forall y_s \in \{1...K_s\}, \quad \mathbb{P}(A_1^{(y_s)}) \geq \mathbb{P}(A_2^{(y_s)}) \geq ... \geq \mathbb{P}(A_{K_m}^{(y_s)});$

2. $\forall y_s \in \{1...K_s\}, \forall i \in \{1...K_m\}, \quad \mathbb{P}(A_i^{(y_s)}) = \mathbb{P}(B_i^{(y_s)}).$

Let us now define $C^{(y_s)}$, the set of partitions of $\Omega$ whose elements have the same probabilities as $(A_i^{(y_s)})_{i \in \{1...K_m\}}$. That is,

$$
C^{(y_s)} = \{(U_i)_{i \in \{1...K_m\}} \ / \ \bigcup_i U_i = \Omega; \quad \forall(i,j), i \neq j, U_i \cap U_j = \varnothing; \quad \forall i, \mathbb{P}(U_i) = \mathbb{P}(A_i^{(y_s)})\}.
\tag{15}
$$

It is clear that: $(B_i^{(y_s)})_{i \in \{1...K_m\}} \in C^{(y_s)}$.

Hence, it is also clear that:

$$
\sum_{y_s} \mathbb{P}(Y_s' = y_s) \sum_i \mathbb{P}(A_i^{(y_s)} \cap B_i^{(y_s)}) \geq \sum_{y_s} \mathbb{P}(Y_s' = y_s) \min_{(U_i^{(y_s)}) \in C^{(y_s)}} \sum_i \mathbb{P}(A_i^{(y_s)} \cap U_i^{(y_s)})
\tag{16}
$$

Therefore, we can lower bound the accuracy in equation (14) using the inequality (16) above, such that:

$$
\overbrace{\mathbb{P}(\pi_m(Z') = Y_m')}^{\text{Accuracy}} \geq \sum_{y_s} \mathbb{P}(Y_s' = y_s) \min_{(U_i^{(y_s)}) \in C^{(y_s)}} \sum_i \mathbb{P}(A_i^{(y_s)} \cap U_i^{(y_s)})
\tag{17}
$$

Let us now separate two cases:

1. $K_m > 2$ and $\mathbb{P}(A_1^{(y_s)}) < 1/2;$

2. $K_m \leq 2$ or $\mathbb{P}(A_1^{(y_s)}) \geq 1/2.$

We shall henceforth ignore the index $(y_s)$ for better clarity.

In case 1, we simply use that:

$$
\forall (U_i^{(y_s)})_{i \in \{1...K_m\}} \in C^{(y_s)}, \sum_i \mathbb{P}(A_i^{(y_s)} \cap U_i^{(y_s)}) \geq 0.
\tag{18}
$$

In case 2, we show that (cf. Lemma 1):

$$
\min_{(B_i)_{i \in \{1...K_m\}} \in C} \sum_i \mathbb{P}(B_i \cap A_i) \geq 2 \times P(A_1) - 1.
\tag{19}
$$

The final result comes from the fact that:

$$\mathbb{P}(A_1) < 1/2 \implies P(A_1) - 1/2 < 0$$

Hence the two cases are summarized by the formula:

$$\min_{(B_i)_{i \in \{1 \ldots K_m\}} \in C} \sum_i \mathbb{P}(B_i \cap A_i) \geq \max \left\{ 0, 2 \left( \max_i \mathbb{P}(A_i) - \frac{1}{2} \right) \right\}.$$

Finally, as the joint laws are assumed equal in distribution, namely $(Y_m, Y_s) \overset{d}{=} (Y_m', Y_s')$, it comes that:

$$
\begin{aligned}
P(A_1^{(y_s)}) &= \max_{y_m} \mathbb{P}(Y_m' = y_m \mid Y_s' = y_s) \\
&= \max_{y_m} \mathbb{P}(Y_m = y_m \mid Y_s = y_s).
\end{aligned}
\tag{20}
$$

$\square$

## A.2  Proof of Theorem 2

**Lemma 2.** *If $Y \perp\!\!\!\perp X \mid Z$ and $X, Y, Z$ are discrete random variables then,*

$$\forall (x, y, z) \in (X(\Omega) \times Y(\Omega) \times Z(\Omega)) \quad \text{with} \quad \mathbb{P}(X = x \cap Z = z) > 0,$$
$$\mathbb{P}(Y = y \mid X = x, Z = z) = \mathbb{P}(Y = y \mid Z = z)$$

*Proof.* $\forall (x, y, z) \in (X(\Omega) \times Y(\Omega) \times Z(\Omega))$ such that $\mathbb{P}(X = x \cap Z = z) > 0$ :

$$
\begin{aligned}
\mathbb{P}(Y = y \mid X = x, Z = z) &= \frac{\mathbb{P}(X = x, Y = y, Z = z)}{\mathbb{P}(X = x, Z = z)} \\
&= \frac{\overbrace{\mathbb{P}(X = x \mid Y = y, Z = z)}^{\text{cf. Assumption}} \mathbb{P}(Y = y, Z = z)}{\mathbb{P}(X = x, Z = z)} \\
&= \frac{\mathbb{P}(X = x \mid Z = z) \mathbb{P}(Y = y, Z = z)}{\mathbb{P}(X = x, Z = z)} \\
&= \frac{\mathbb{P}(X = x \mid Z = z) \mathbb{P}(Y = y \mid Z = z)}{\mathbb{P}(X = x \mid Z = z)} \\
\mathbb{P}(Y = y \mid X = x, Z = z) &= \mathbb{P}(Y = y \mid Z = z)
\end{aligned}
$$

$\square$

**Theorem 2.** *If we assume that:*

- $Z \overset{d}{=} Z'$;
- $(Z \mid Y_s = k) \overset{d}{=} (Z' \mid Y_s' = k), \forall k \in \{1, ..., K\}$;
- $Z' \perp\!\!\!\perp Y_m' \mid Y_s'$;

*then the accuracy of the model is:*

$$\mathbb{P}(\pi_m(Z') = Y_m') = \sum_{y_s} \left[ \mathbb{P}(Y_s = y_s) \sum_{y_m} \mathbb{P}(Y_m = y_m \mid Y_s = y_s)^2 \right].
\tag{21}$$

*Proof.* Let $(\Omega, \mathcal{F}, \mathbb{P})$ be a probability space. Let $Y_m(\Omega) = Y_m'(\Omega) = \{1, ..., K_m\}$ and $Y_s(\Omega) = Y_s'(\Omega) = \{1, ..., K_s\}$. In the following, for the sake of clarity we shall try to omit writing $Y_s(\Omega)$ and $Y_m(\Omega)$. Thus, when no confusion is possible we shall write $\bigcup_{y_s}$ instead of $\bigcup_{y_s \in Y(\Omega)}$ .

Using the law of total probability, with $\bigcup_{y_s}\{Y'_s = y_s\} = \Omega$, it comes that:

$$\overbrace{\mathbb{P}(\pi_m(Z') = Y'_m)}^{\text{Accuracy}} = \sum_{y_s} \underbrace{\mathbb{P}(\pi_m(Z') = Y'_m \mid Y'_s = y_s)}_{A^{(y_s)}} \mathbb{P}(Y'_s = y_s). \tag{22}$$

Similarly, we reformulate $A^{(y_s)}$ with the law of total probability, using that $\bigcup_{y_m}\{Y'_m = y_m \mid Y'_s = y_s\} = \Omega$, and it comes that:

$$A^{(y_s)} = \sum_{y_m} \mathbb{P}(\pi_m(Z') = y_m \mid Y'_s = y_s, Y_m{}' = y_m)\mathbb{P}(Y_m{}' = y_m \mid Y'_s = y_s).$$

We now replace $A^{(y_s)}$ in equation 22, which is the accuracy, and it comes that:

$$\overbrace{\mathbb{P}(\pi_m(Z') = Y'_m)}^{\text{Accuracy}} = \sum_{y_s} \mathbb{P}(Y'_s = y_s) \sum_{y_m} \mathbb{P}(\pi_m(Z') = y_m \mid Y'_s = y_s, Y_m{}' = y_m)\mathbb{P}(Y_m{}' = y_m \mid Y'_s = y_s). \tag{23}$$

If $Y'_m \perp\!\!\!\perp Z' \mid Y'_s$ (assumption 3), it can easily be shown (cf. Lemma 2) for all $(y_s, y_m)$ such that $\mathbb{P}(Y'_s = y_s, Y'_m = y_m) > 0$, we have:

$$\mathbb{P}(\pi_m(Z') = y_m \mid Y'_s = y_s, Y_m{}' = y_m) = \mathbb{P}(\pi_m(Z') = y_m \mid Y'_s = y_s) \tag{24}$$

Furthermore, it is clear that

$$\mathbb{P}(Y'_s = y_s, Y'_m = y_m) = 0 \implies \mathbb{P}(Y'_s = y_s \mid Y'_m = y_m) = 0.$$

Hence, we can rewrite equation 23, namely the accuracy, using equation 24 for all $(y_s, y_m)$, and it comes that:

$$\overbrace{\mathbb{P}(\pi_m(Z') = Y_m{}')}^{\text{Accuracy}} = \sum_{y_s} \mathbb{P}(Y'_s = y_s) \sum_{y_m} \mathbb{P}(\pi_m(Z') = y_m \mid Y'_s = y_s)\mathbb{P}(Y_m{}' = y_m \mid Y'_s = y_s). \tag{25}$$

From assumption 2 on conditional features alignment, namely $\forall k \in Y_s(\Omega), (Z \mid Y_s = k) \overset{d}{=} (Z' \mid Y'_s = k)$, and given that the classifier $\pi_m$ is fixed, it comes that:

$$\forall(y_m, y_s), \quad \mathbb{P}(\pi_m(Z') = y_m \mid Y'_s = y_s) = \mathbb{P}(\pi_m(Z) = y_m \mid Y_s = y_s). \tag{26}$$

We assumed that the classifier $\pi_m$ is perfect on the training set, such that:

$$\forall(y_m, y_s), \quad \mathbb{P}(\pi_m(Z) = y_m \mid Y_s = y_s) = \mathbb{P}(Y_m = y_m \mid Y_s = y_s). \tag{27}$$

Hence, combining equality 27 and equality 26, it comes that:

$$\forall(y_m, y_s), \quad \mathbb{P}(\pi_m(Z') = y_m \mid Y'_s = y_s) = \mathbb{P}(Y_m = y_m \mid Y_s = y_s). \tag{28}$$

We now rewrite the accuracy, that is equation 25, using the equality 28 above, and it comes that:

$$\overbrace{\mathbb{P}(\pi_m(Z') = Y_m{}')}^{\text{Accuracy}} = \sum_{y_s} \mathbb{P}(Y'_s = y_s) \sum_{y_m} \mathbb{P}(Y_m = y_m \mid Y_s = y_s) \underbrace{\mathbb{P}(Y_m{}' = y_m \mid Y'_s = y_s)}_{\text{Joint distributions of labels}}. \tag{29}$$

We assumed that the joint distributions of the labels were constant over time, i.e., $(Y_m \cap Y_s) \overset{d}{=} (Y_m' \cap Y_s')$. Consequently, we replace the test time joint distribution by their training counterpart in equation 29, such that:

$$\overbrace{\mathbb{P}(\pi_m(Z') = Y_m')}^{\text{Accuracy}} = \sum_{y_s} \underbrace{\mathbb{P}(Y_s' = y_s)}_{\text{Prior distribution}} \sum_{y_m} \mathbb{P}(Y_m = y_m \mid Y_s = y_s) \mathbb{P}(Y_m = y_m \mid Y_s = y_s). \quad (30)$$

Finally, we assumed that the prior distributions of the labels are constant over time, i.e., $Y_s \overset{d}{=} Y_s'$ Therefore, we replace the test time prior by the training time prior in equation 30 and it gives:

$$\mathbb{P}(\pi_m(Z') = Y_m') = \sum_{y_s} \mathbb{P}(Y_s = y_s) \sum_{y_m} \mathbb{P}(Y_m = y_m \mid Y_s = y_s)^2. \quad (31)$$

$\square$

## B  Implementation Details

**Joint Training.**  We use the same hyper-parameters as [44] to train the ResNet-50 on the classification and contrastive tasks jointly. We set the batch size to 256 and the weight of the self-supervised task $\lambda$ to 0.1 in all experiments. We train the model for 1,000 epochs on CIFAR-10 and CIFAR-100 from scratch. On VisDA, we reduce the number of epochs to 100 and warm start the training from a pre-trained ResNet-50 due to limited training data.

**Test-Time Adaptation.**  At test-time, we adapt the encoder using stochastic gradient descent with a learning rate of 0.001 and momentum of 0.9. We use a batch size of 256 for the self-supervised task and online feature alignment. Our experiments are conducted on GeForce RTX 3090.

**Contrastive Task.**  We use the same data augmentation strategy as [12]. For random cropping, we first create crops of random size and aspect ratio from raw images and subsequently resize them to the original size. For color distortion, we set the strength of color jitter to 0.5. We set the temperature parameter to 0.5 for CIFAR-10, CIFAR10-C, CIFAR-100 and CIFAR100-C, and 0.1 for the VisDA dataset.

## C  Additional Experiments

### C.1  Additional Results on Common Corruption Datasets

In addition to the bar plot in Figure 3 from the main paper, we summarize the classification errors on CIFAR10-C with different severity levels of corruptions in Tables C.1-C.3. Across all three levels, our proposed TTT++ outperforms other strong baselines [8, 35, 36] by a clear margin. Specifically, our method leads to ∼23% lower classification errors on average than prior state-of-the-art methods.

### C.2  Additional Results with Different Random Seeds

We follow the evaluation protocol of previous work [6, 8] and run all methods on the same pre-trained model with the same seed. As shown in Table C.4, the variance across different random seeds is minimal. We therefore report our main experimental results with only one random seed.

### C.3  Additional Qualitative Results

In addition to Figure 3 from the main paper, we visualize the learned representation of test images on three other types of corruption in Figures C.1. These qualitative results confirm that while TTT-C itself leads to semantically more separated feature clusters, it cannot resolve the distributional shifts in the feature space. In comparison, the full version of our proposed TTT++ is able to improve both the feature alignment and the discriminative power of the test-time representations simultaneously.

Table C.1: Classification error (%) on CIFAR10-C, level-5 corruptions.

| | brit | contr | defoc | elast | fog | frost | gauss | glass | impul | jpeg | motn | pixel | shot | snow | zoom | Average |
|---|---|---|---|---|---|---|---|---|---|---|---|---|---|---|---|---|
| Test | 7.01 | 13.27 | 11.84 | 23.38 | 29.41 | 28.24 | 48.73 | 50.78 | 57 | 19.46 | 23.38 | 47.88 | 44 | 21.93 | 10.84 | 29.14 |
| BN [35] | 8.22 | 8.27 | 9.66 | 19.54 | 19.95 | 19.5 | 17.11 | 25.95 | 27.7 | 13.67 | 13.72 | 11.50 | 16.17 | 15.88 | 7.93 | 15.65 |
| TENT[8] | 7.14 | 7.16 | 8.28 | 16.86 | 14.49 | 11.99 | 14.64 | 21.39 | 22.1 | 12.01 | 11.28 | 9.6 | 13.34 | 12.16 | 7.15 | 12.64 |
| SHOT [36] | 8.01 | 7.95 | 9.51 | 18.93 | 18.88 | 13.15 | 16.42 | 24.74 | 26.27 | 13.55 | 13.39 | 11.23 | 15.38 | 15.55 | 7.74 | 14.71 |
| TFA | 7.44 | 7.40 | 8.89 | 15.73 | 12.82 | 11.49 | 12.94 | 18.46 | 19.13 | 11.66 | 10.77 | 9.93 | 12.67 | 11.73 | 7.03 | 11.87 |
| TTT-C | 5.32 | 5.7 | 8.05 | 15.37 | 8.39 | 11.11 | 14.63 | 19.87 | 12.41 | **9.54** | 8.76 | 11.93 | 13.06 | 9.91 | 7.1 | 10.74 |
| TTT++ | **5.20** | **5.43** | **7.73** | **13.08** | **8.09** | **9.73** | **12.73** | **15.70** | **12.45** | 10.39 | **8.52** | **8.87** | **11.07** | **8.75** | **6.31** | **9.60** |

Table C.2: Classification error (%) on CIFAR10-C, level-4 corruptions.

| | brit | contr | defoc | elast | fog | frost | gauss | glass | impul | jpeg | motn | pixel | shot | snow | zoom | Average |
|---|---|---|---|---|---|---|---|---|---|---|---|---|---|---|---|---|
| Test | 5.88 | 7.45 | 8.32 | 13.04 | 13.02 | 20.07 | 43.31 | 52.34 | 43.78 | 17.12 | 16.72 | 26.45 | 34.34 | 19.31 | 8.12 | 21.95 |
| BN [35] | 7.33 | 7.48 | 8.19 | 13.46 | 13.10 | 11.50 | 15.63 | 25.36 | 21.65 | 12.11 | 12.35 | 8.98 | 12.91 | 16.70 | 7.05 | 12.92 |
| TENT [8] | 6.71 | 6.62 | 7.08 | 11.73 | 9.13 | 10.66 | 13.61 | 20.39 | 17.12 | 10.77 | 10.02 | 8.56 | 11.04 | 13.41 | 6.59 | 10.90 |
| SHOT [36] | 6.71 | 6.90 | 7.66 | 12.31 | 11.22 | 10.77 | 14.30 | 22.49 | 18.68 | 11.33 | 11.13 | 8.51 | 11.58 | 15.05 | 6.68 | 11.69 |
| TFA | 6.55 | 6.51 | 7.38 | 11.76 | 9.96 | 10.03 | 12.65 | 18.46 | 15.39 | 10.45 | 10.36 | 8.36 | 10.69 | 12.79 | 6.47 | 10.52 |
| TTT-C | 4.85 | 5.02 | 6.14 | 10.17 | 6.00 | 8.47 | 12.84 | 19.90 | 11.48 | 10.58 | 8.17 | 7.43 | 10.24 | 10.44 | 6.15 | 9.19 |
| TTT++ | **4.34** | **4.81** | **5.68** | **9.52** | **5.91** | **7.74** | **12.08** | **15.92** | **9.47** | **9.34** | **7.71** | **6.93** | **9.26** | **9.08** | **5.80** | **8.24** |

Table C.3: Classification error (%) on CIFAR10-C, level-3 corruptions.

| | brit | contr | defoc | elast | fog | frost | gauss | glass | impul | jpeg | motn | pixel | shot | snow | zoom | Average |
|---|---|---|---|---|---|---|---|---|---|---|---|---|---|---|---|---|
| Test | 5.64 | 6.47 | 5.73 | 7.69 | 8.98 | 18.54 | 36.96 | 35.53 | 26.86 | 15.54 | 16.68 | 13.10 | 28.00 | 16.89 | 7.54 | 16.68 |
| BN [35] | 6.95 | 6.96 | 7.03 | 9.27 | 10.19 | 11.21 | 13.53 | 16.53 | 15.84 | 10.91 | 12.20 | 8.42 | 12.12 | 14.90 | 7.26 | 10.89 |
| TENT [8] | 6.51 | 6.44 | 6.36 | 8.63 | 7.90 | 9.87 | 11.88 | 14.26 | 12.99 | 10.38 | 10.58 | 7.24 | 9.97 | 11.87 | 6.67 | 9.44 |
| SHOT [36] | 6.58 | 6.66 | 6.80 | 8.67 | 9.12 | 10.46 | 12.14 | 15.17 | 14.06 | 10.40 | 10.93 | 7.74 | 10.72 | 12.78 | 6.59 | 9.92 |
| TFA | 6.32 | 6.46 | 6.63 | 8.61 | 8.78 | 10.17 | 11.10 | 13.23 | 11.54 | 9.99 | 10.20 | 7.49 | 10.21 | 12.03 | 6.70 | 9.30 |
| TTT-C | 4.51 | 4.81 | 4.77 | 6.79 | 5.34 | 8.99 | 11.38 | 12.93 | 8.63 | 9.86 | 8.09 | 6.49 | 9.49 | 8.70 | 5.95 | 7.78 |
| TTT++ | **4.26** | **4.50** | **4.68** | **6.47** | **5.18** | **7.84** | **9.92** | **10.99** | **8.06** | **8.51** | **7.66** | **5.97** | **8.43** | **7.78** | **5.46** | **7.05** |

Table C.4: Classification error (%) of TTT+ with different random seeds on CIFAR10-C, level-5 corruptions.

| Seed | brit | contr | defoc | elast | fog | frost | gauss | glass | impul | jpeg | motn | pixel | shot | snow | zoom | Average |
|---|---|---|---|---|---|---|---|---|---|---|---|---|---|---|---|---|
| 0 | 5.20 | 5.43 | 7.73 | 13.08 | 8.09 | 9.73 | 12.73 | 15.70 | 12.45 | 10.39 | 8.52 | 8.87 | 11.07 | 8.75 | 6.31 | 9.60 |
| 1 | 5.09 | 5.37 | 7.47 | 12.62 | 7.95 | 9.44 | 12.63 | 16.19 | 12.25 | 10.40 | 8.59 | 8.51 | 11.22 | 8.71 | 6.12 | 9.50 |
| 2 | 5.25 | 5.50 | 7.69 | 13.04 | 8.17 | 9.46 | 13.05 | 16.21 | 11.95 | 10.49 | 8.57 | 8.48 | 11.14 | 8.76 | 6.34 | 9.61 |
| Std | 0.08 | 0.07 | 0.14 | 0.25 | 0.11 | 0.16 | 0.22 | 0.29 | 0.25 | 0.06 | 0.04 | 0.22 | 0.08 | 0.03 | 0.12 | 0.06 |

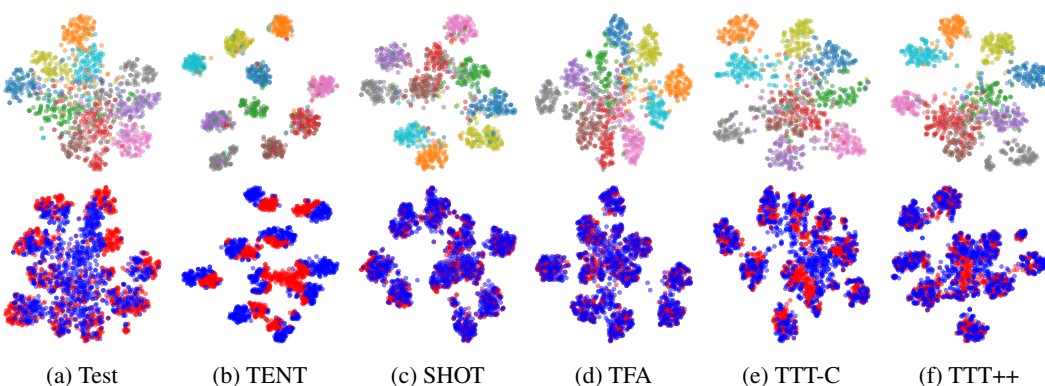

| (a) Test | (b) TENT | (c) SHOT | (d) TFA | (e) TTT-C | (f) TTT++ |

Figure C.1: T-SNE visualization of the representation for the CIFAR10 images with the level-5 elastic transform corruption. Top row: per-class feature distribution. Bottom row: marginal feature distribution on the original test images (red) and corrupted test images (blue).