# OpenReview forum: "TTT++: When Does Self-Supervised Test-Time Training Fail or Thrive?"
_NeurIPS.cc/2021/Conference — NeurIPS 2021 Poster_

### Official Review · Reviewer_qJ8x · 2021-07-15

**Rating:** 6
**Confidence:** 3

**Summary:**

This paper is about test-time adaptation (TTA), in particular about improving the recently published test-time training (TTT) approach. It provides an analysis about conditions that can lead to TTT failures, and overcome them with a method that relies on i) feature alignment and ii) more sophisticated self-supervised learning (SSL) objectives (when compared with the TTT's seminal work). The method is compared against other recent TTA approaches, showing good performance on standardized benchmarks.

**Limitations And Societal Impact:**

Yes.

**Main Review:**

This is a well written paper, concerning an important, recently introduced topic. The proposed method is properly located in the literature, and the ideas are well argumented and presented. Furthermore, the proposed method is very reasonable.

TTA has the potential to revolutionize the deployment of machine learning systems in the real world, but only a bunch of algorithms exists as of today. Improving one of the leading algorithms in this niche topic (TTT, by Sun et al. [8]), this paper brings valuable contributions to the table; indeed, while effective, TTT relies on an SSL approach that is currently far from the state of the art: in this work, the importance of using more effective SSL objectives is shown, and some more methodological contributions are explored.

To improve the paper, I believe more ablation studies will allow to properly assess the efficacy of the proposed strategy. In particular:

1) One of the powerful facet of TTA is the possibility to adapt on the fly to single test samples, or at least to a few samples. In this regard, I could not understand in which of the following two ways test-time adaptation is performed: A) adapt on all test samples via SGD (with the indicated batch size) and at the end evaluate performance on all test samples, or B) adapt on a batch (with the indicated batch size) and evaluate the model on that batch (if this is the case, the adaptation process starts from scratch at every step, of from the point reached by adapting to the previous batch?). I believe that the setting B) is the most interesting for TTA, because, as mentioned above, it would allow adapting on-the-fly to single samples. In my opinion, this work should address this point, by assessing how performance changes - also with respect to competing algorithms - as the batch size varies (in my understanding, in this work the batch size at test time is kept fixed - see Sec. C). For a reference, see the analyses performed in Schneider et al. [42].

2) Related to point 1), I believe that some more ablation studies to understand the trade-off between the hyper-parameters in Eq.4 should be provided. Related to this, I found curious to see in Tables D.1 and D.2 that using the covariance matrix alone helps more than using the mean vectors alone: the covariance matrix alone only carries information about how feature vectors are spread in their space, not about their location. What would happen without the SSL objective (that is, by setting $\lambda_s=0$ in Eq.4)?

3) Minor - I did not find the analysis in Sec. 3.1. very enlightening, because the main task and the SSL task are very artificial. An example where it is shown that the proposed method avoids degenerate solutions would be significantly more helpful, in my opinion.

Note: I am not able to properly assess the usefulness and the correctness of the theoretical results in Sec. 4.1 (as well as their proofs), since they are beyond my expertize.

I believe this is a good paper, with some parts still lacking a bit of work. I look forward reading the Author response, discussing with other Reviewers and updating my score accordingly.

**Time Spent Reviewing:**

/

---

> ### Author Response · Authors · 2021-08-10
> **Response to Reviewer qJ8x**
>
> Thank you for the detailed review and thoughtful feedback. Below we answer specific questions.
>
> >**Why does covariance matrix alone help more than using the mean vectors alone?**
>
> - The covariance matrix captures both the spread of each feature component and the correlation between different components, whereas the mean vector is only about the location in the space.
> - As shown in Fig.D1-D2, some common domain shifts tend to *shrink* the separation between feature clusters. This effect does not change the central location much, but it can dramatically distort the shape of the feature distribution. As such, matching the covariance matrix alone is often more effective than matching the mean vector alone.
>
> >**What would happen without the SSL objective (that is, by setting $\lambda=0$ in Eq.4)?**
>
> - Table D.2 (supplementary material) reports the results of online feature alignment using different moments *without the SSL objective*. The results confirm the advantage of matching both the first and second-order moments over only one of them.
>
> >**An example where it is shown that the proposed method avoids degenerate solutions would be helpful**
>
> - Thanks for the suggestion. We will update the motivation example, as described in the general response.
>
> >**In which setting the test-time adaptation is performed? How does performance change as the batch size varies?**
>
> - We followed the experimental setting of TENT, in which the source model is adapted from test examples batch-by-batch and evaluated on the entire test set. To our knowledge, the test-time algorithms that perform well in this setting are typically also able to adapt well on-the-fly.
> - As suggested, we compare all methods given a variable number of test samples for adaptation (same as the setup of Fig.1 in Schneider et al.). The result below (CIFAR-10 with the level-5 snow corruption) shows that our proposed TTT++ is particularly advantageous when there are 64 or more samples available, while remaining competitive in the low data regime.
>
> 	| # Samples |   16  |   32  |   64  |  128  |  256  |  512  |
> 	|:--------:|:-----:|:-----:|:-----:|:-----:|:-----:|:-----:|
> 	|    BN    | 19.03 | 18.08 | 17.23 | 16.69 | 16.56 | 16.37 |
> 	|   TENT   | 15.99 | 15.67 | 15.48 | 15.32 | 15.03 | 14.72 |
> 	|   SHOT   | 17.97 | 17.25 | 16.83 |  16.5 | 16.17 | 16.09 |
> 	|   TTT-C  | 18.96 | 17.52 | 16.52 | 15.72 | 14.97 | 14.04 |
> 	| TTT++ (Ours) | 16.53 | 15.64 | 14.42 | 14.34 | 13.56 | 13.11 |
>
> >**Some more ablation studies to understand the trade-off between the hyper-parameters in Eq.4**
>
> - Thanks for the suggestion. The table below reports the results of the proposed TTT++ using different $\lambda$ on CIFAR10 with the level-5 snow corruption. We scale the default $\lambda$ parameter by a factor ranging from 0.05 to 50. Note that the results at the two ends of the scaling range are close to that from TTT-C and TFA respectively (in Table D.3).
>
> 	|   $\lambda$ Scale  |    0.05   |    0.1    |    0.5   |   1.0    |   5.0    |   10.0    |   50.0    |
> 	|:------------:|:----------:|:---------:|:---------:|:--------:|:--------:|:---------:|:----------:|
> 	|     TTT++    |   10.11  |   10.93  |   9.82   |   9.08   |   9.06   |  10.55 | 11.21 |

---

> > ### Comment · Reviewer_qJ8x · 2021-08-19
> > **Thank you for your comments and additional experiments**
> >
> > Regarding the experimental protocol ("the source model is adapted from test examples batch-by-batch and evaluated on the entire test set"), do you mean the "online" setup of TENT? In TENT's original paper, two setups are evaluated: online and offline. From their Section 3.1, "For online adaptation, no termination is necessary, and iteration continues as long as there is test data. For offline adaptation, the model is first updated and then inference is repeated. Adaptation may of course continue by updating for multiple epochs".
> >
> > Can the proposed method successfully handle the two different setups? The offline one ('B' in my review) is important, too, because one does not always have large test sets to process all together - sometimes we have single or few samples. The results provided in response to my review seem to suggest that the proposed method may not outperform TENT in such settings, due to comparable performance with 16 test samples: in the offline setup, each adaptation takes into account one or few test samples, hence we are in similar conditions.
> >
> > I believe this is an important point to be addressed: I would still consider the method valuable even if it did not outperform TENT in all settings, but more analyses should be provided in the manuscript.
> >
> > As of now, I would like to maintain my original rating (6) on this paper: I have a positive view overall, but I do not believe it is mature enough to fully recommend acceptance. I will be happy to discuss more with Authors and Reviewers.

---

> > > ### Author Response · Authors · 2021-08-20
> > > **Thanks for the comments**
> > >
> > > Thanks for the detailed comments!
> > >
> > > * We'd like to clarify that the experimental comparison reported in the TENT paper is mainly in the offline setting (same as the setting 'A' in your review), where the entire test set can be used for adaptation for multiple epochs. Please refer to their description ``adaptation is offline for fair comparison`` in both Section 4.1 and 4.2. We follow this experimental setting since all recent test-time methods fit in.
> > >
> > > * We agree that the described adaptation setting B (only one or few samples are available on-the-fly) is appealing and similar to the setup of small batch sizes, where our method performs on par with TENT and better than other baselines.
> > >
> > > * In addition to the discussions above, the **most critical challenge for test-time adaptation**, in our view, is not yet in the online few-shot scenario (since in most cases ML models can presumably wait until receiving enough unlabeled examples from the new domain), but rather to ensure the robustness of adaptation under various kinds of distributional shifts.
> > >   * In this regard, all previous methods have clear shortcomings, as evidenced in our results (Table 1, 2, D) as well as the open issues in the TENT repo.
> > >   * In comparison, our method is (i) empirically advantageous (or at the very least competitive) under both common corruptions and natural shifts (ii) theoretically grounded with a lower bound on the test-time accuracy (iii) algorithmically insightful, suggesting the importance of bringing extra information to the test time, namely task-specific information through strong SSL and model-specific information through feature summarization.
> > >
> > > We'll be delighted to discuss further with Reviewers.

---

> > > > ### Comment · Reviewer_qJ8x · 2021-08-23
> > > > **Thank you for further explanations**
> > > >
> > > > Thank you for clarifying. I made some confusion with online/offline definitions in my comments - I actually had misinterpreted what Wang et al. meant by "offline" adaptation.
> > > >
> > > > I will further discuss with ACs and Reviewers to finalize the process. I am not fully recommending acceptance myself because I still believe that the setting where the method is comparable with TENT is a crucial one, as well as the online one. More analyses in this regard will make the work significantly more robust.
> > > >
> > > > That said, while the rating does not reflect it, I am more convinced by this paper than before discussing with the Authors and reading the discussion with Reviewer qJr7. I would have no objections to accepting it, if this were the final decision.

---

> > > > > ### Author Response · Authors · 2021-08-31
> > > > > **Response to Reviewer qJ8x's Additional Comments**
> > > > >
> > > > > Thanks for the comments! We will be glad to look more into the online setting in our future work.

---

### Official Review · Reviewer_qJr7 · 2021-07-16

**Rating:** 6
**Confidence:** 5

**Summary:**

This paper focus on improving the generalization of deep neural networks in the presence of distributional shift at test time i.e., unsupervised source-free network adaptation on data distribution that is different from training. They follow the footsteps of a prior work by Sun et al. [8] and leverage an auxiliary task at test time. However, this work points out that test time training with auxiliary tasks like rotation prediction need not necessarily encourage the network to improve its generalization on the main task. Thus, the authors propose to transfer the benefits of self-supervised learning (SSL) to the main task by introducing an online feature alignment strategy that utilizes offline feature summarization statistics collected on the original training (source) data and also choosing more suitable SSL task like contrastive learning instead of rotation prediction. Tthey denote the resulting approach as TTT++. Results and comparison to existing techniques are shown on popular datasets like cifar10-c, cifar100-c, cifar10.1 and VisDA-C.

**Limitations And Societal Impact:**

I appreciate the authors for addressing the limitations and societal impact sufficiently.

**Main Review:**

## Originality ##
The online feature alignment strategy is inspired by distribution matching measures from the domain adaptation literature, however proposed firstly in this work for test time adaptation. Additionally, the authors employ the popular contrastive task as the SSL task. It is clearly distinguished from the previous works and sufficiently cited. In general, this work introduces concepts  from other fields into test-time adaptation, that are not novel in general. Originality is thus relatively small.

## Quality ##
Technical details of this work are clear. They provide theoretical analysis about the relationship between SSL and main task. The method is tested on popular benchmarks. This is a complete piece of work. Authors are honest about strengths and weaknesses.

The authors motivate the online feature alignment strategy by stating that simply applying SSL based adaptation results in poor accuracy. However, results on cifar100-c and cifar10.1 show that such strategy included in TTT++ is almost comparable to network adaptation with contrastive based SSL task and slightly better for cifar10-c and little prominent on VisDA-C. The improvements from such alignment strategy on cifar10-c and VisDA-C, but not on cifar100-c raises a question whether this technique scales to dataset with larger number of classes (e.g. ImageNet). Overall, results suggest that the impact of the online feature alignment strategy is not as pronounced as strong as authors claim.

## Clarity ##
The paper is nicely written, well organized and easy to read.

Comments:
* Section 3.1 provides an illustrative example of a failure case. Though this section is easy to understand, it is not sufficient to reason about a failure case. A synthetic problem is provided in appendix. I suggest authors may consider moving parts of appendix to main paper to provide sufficient rationale on the argument.
* Figure 4 illustrates the encoder feature distribution. Despite authors state that TTT+ displays improved per-class separation, it is not clear to infer this conclusion from the figure. It appears that SHOT and TTT++ show similar per-class feature distribution and feature alignment. I suggest the authors to consider a different illustration or representation to emphasize their claim from Figure 4, ideally provide quantitative numbers rather than only a visualization.

## Significance ##
The results indicate that the proposed approach outperforms previous works on cifar10-c, cifar100-c and cifar10.1. However, the major improvements come from replacing the rotation prediction task from Sun et al. [8] with the contrastive learning and this choice of SSL task seems straightforward given its recent popularity.  The improvements from the online feature alignment are relatively small and removing it would not result in "Test-Time Training Failing"; this theme from Section 3 is actually not observed anywhere in the results.

Moreover, the numbers reported from SHOT in Table 2 are not consistent with the results from the original paper Liang et al. [9]. The per-class error rate is reported as 17.1% on VisDA-C in Liang et al. [9], whereas it is reported as 25.04% in the Table 2 comparisons. Taking the error rate reported in original paper into account, TTT++ does not improve upon SHOT[9].

It is not clear how the hyperparameters are chosen for the test time adaptation. Are they chosen based on the performance on the test corruptions of the respective benchmark or the validation corruptions? Do the hyperparameters vary across the corruptions?

I appreciate the results obtained with contrastive learning as an auxiliary task that outperforms the previous works. However, based on my above comments, I see this as a simple extension of Sun et al. [8] with small novelty. I consider this submission as marginally below acceptance threshold and would be willing to reconsider my score based on the author responses and other reviews.

## Update after author response ##
The authors have addressed my comments and questions and I am raising my score to 6, that is: I see the paper slightly above the acceptance threshold.
The main limitations that prevent me from going to a score of 7 or higher is that (a) the proposed feature alignment strategy in its current form is limited to problems with few classes and (b) most (but not all) performance gains compared to prior work TTT come from using a stronger and well established self-supervised contrastive learning objective.


**Time Spent Reviewing:**

4

---

> ### Author Response · Authors · 2021-08-10
> **Response to Reviewer qJr7**
>
> Thank you for the detailed review and thoughtful feedback. Below we answer specific questions.
>
> >**The numbers reported from SHOT in Table 2 are not consistent with the results from the original paper Liang et al. [9]**
>
> - This difference arises from the disparity of backbone architectures. As described in Line 261, ``we use ResNet-50 as the backbone architecture throughout our experiments``, whereas the original SHOT paper used ResNet-101. As such, the test error of the source model *without any adaptation* in their paper (53.4%) is also lower than the results we obtained with ResNet-50 (58.7%) on VisDA-C.
> - Note that the choice of backbone models is often inconsistent in recent test-time literature (ResNet-18, ResNet-26, ResNet-50, ResNet-101, ResNeXt, etc.). For a fair comparison, we used ResNet-50 for all methods in our experiments.
> - Under this setup, the result we obtained and reported for TENT is actually better than that from the original paper, as summarized below.
>
> 	| Dataset    |  Original Paper | Our Implementation |
> 	|:----------:|:---------------:|:------------:|
> 	| CIFAR10-C  |       14.3      |      12.6    |
> 	| CIFAR100-C |       37.3      |      36.3    |
>
> >**Whether the alignment technique scales to datasets with a larger number of classes?**
>
> - Scaling to a large number of classes is a common challenge to feature alignment techniques (cf. the effect of category number in [1]). As discussed in L312-313, a good estimate of moments often requires a handful of samples per class. As such, our current feature alignment strategy needs a large batch size for ImageNet-C.
> - However, we would like to highlight that the proposed moment matching is the very first instance of our online feature alignment framework. Modifications on top of the current strategy are expected to scale better and can be explored in future work, e.g., decoupling the sample size from batch size [2] or employing other forms of feature summarization [3].
>
> >**I suggest authors move the synthetic problem (appendix) to provide sufficient rationale**
>
> - Thanks for the suggestion! We will update the synthetic problem and move it to Figure 1, as described in the general response above.
>
> >**Quantitative numbers for per-class feature alignment**
>
> - Thanks for the suggestion! To complement the t-sne visualization, we compute the Sinkhorn divergence [3,4] between the training and test class-conditional feature distributions. The results are averaged across all classes.
> - The quantitative results below show that while SHOT and TTT++ lead to visually similar t-sne, the class-conditional divergence from TTT++ is smaller thanks to the discriminative power gained from the strong self-supervised task. They also confirm our observations in Figure 4, D.1 and D.2 that TENT and TTT-C may suffer from significant feature misalignment.
>
> 	| Method |  Fog  | Elastic | Glass |
> 	|:------:|:-----:|:-------:|:-----:|
> 	|  TENT  | 12.96 |  20.50  | 16.94 |
> 	|  TTT-C |  5.70 |   6.58  |  6.60 |
> 	|  SHOT  |  5.12 |   5.43  |  6.39 |
> 	|  TTT++ (Ours) |  4.35 |   4.69  |  4.95 |
>
> >**"Test-Time Training Failing" is not observed in the results**
>
> - Our analysis in Sec. 3 aims to draw attention to the risk of feature misalignment in test-time training.
> - This risk can be clearly verified under large domain shifts in the controlled synthetic experiment, as described in the general response.
> - On current image robustness benchmarks, the domain shifts are often not as severe. Yet, we can still observe a clear performance gap between the TTT with and without feature alignment. Moreover, the performance gain ($1 - Err_\textbf{TTT++} / Err_\textbf{TTT-C}$) tends to grow along with the rise of image corruption level, as shown in the result below (calculated from Table D.3-5). This result confirms the importance of feature alignment for test-time training in real scenarios.
>
> 	|        Cifar10-C        | Feature Alignment Gain |
> 	|:------------------------:|:-------------------:|
> 	| Level 3 Corruption |   9.4%   |
> 	| Level 4 Corruption |   10.3%  |
> 	| Level 5 Corruption |   11.1%  |
>
> [1] Moment Matching for Multi-Source Domain Adaptation, ICCV’19\
> [2] Momentum Contrast for Unsupervised Visual Representation Learning, CVPR'20\
> [3] Prototypical Contrastive Learning of Unsupervised Representations, ICLR'21\
> [4] Sinkhorn Distances: Lightspeed Computation of Optimal Transport, NeurIPS’13\
> [5] Interpolating between Optimal Transport and MMD using Sinkhorn Divergences, AISTATS’19

---

> > ### Comment · Reviewer_qJr7 · 2021-08-17
> > **Final review**
> >
> > I would like to thank the authors for their response. In general, the authors have addressed my concerns and questions and I have increased my score to "6: Marginally above the acceptance threshold". The main limitations that prevent me from going to a score of 7 or higher is that (a) the proposed feature alignment strategy in its current form is limited to problems with few classes and (b) most (but not all) performance gains compared to prior work (TTT) come from using a stronger and well established self-supervised contrastive learning objective.

---

> > > ### Author Response · Authors · 2021-08-31
> > > **Response to Reviewer qJr7's Additional Review**
> > >
> > > Thanks for the comments! We appreciate the feedback on the limitations of our paper and would like to share that the `decoupling` trick mentioned in our earlier response can help address these limitations, as summarized in our additional general response #2.

---

### Official Review · Reviewer_YjL7 · 2021-07-17

**Rating:** 6
**Confidence:** 4

**Summary:**

Authors propose a test time training method using contrastive loss based SSL (Self Supervised Learning) task and feature alignment to address the domain shift between training and test data. Authors give lower bound for the performance during test under some assumptions.

**Limitations And Societal Impact:**

For limitations please check the main review.

**Main Review:**


1) Authors clearly describe the background of the work and their proposed method. Proposed method is a novel combination of well known techniques. The proposed method improves performance by 2.5% on VisDA-C dataset and around 23% for the CIFAR 10 dataset compared to the baseline.

2) Can authors compare the results on the Imagenet-C dataset as previous papers in the literature have done [1]? If that can not be done, can authors justify why?

3) It is not clear from paper how hyperparameters are chosen for Tent and SHOT baselines? It is possible that the baseline method's hyperparameters won't work for this data if the data processing is different. In that case, was there any grid search performed on baseline methods' hyperparameters?

4) It would be good to have a graph of classification error as test time training processes to understand if performance is increasing gradually as training progresses similar to figure 2 from [1].

5) Assumption that test feature distributions are aligned seems a bit strong. Based on what type of corruption applied on test data, the distribution can change. Can authors provide empirical data to back up this claim?

6) It would be useful to show how the method performs on different domain shifts in test data. Does the method perform comparably l with Elastic Transform domain shift as it does with Fog?

7) In equation 3, authors match the first and second moment of the data. Would it be possible to match all the moments of the distribution instead of only 2 moments like in [2].

8) Fundamentally the approach seems very similar to TTT where only the pretext task has changed from rotation to a new self supervised task. Is there any other fundamental change apart from first and second moment matching of the data?


[1] Sun, Yu, et al. "Test-time training with self-supervision for generalization under distribution shifts." International Conference on Machine Learning. PMLR, 2020.

[2] Blanchard, Gilles, et al. "Domain Generalization by Marginal Transfer Learning." J. Mach. Learn. Res. 22 (2021): 2-1.

**Time Spent Reviewing:**

6

---

> ### Author Response · Authors · 2021-08-10
> **Response to Reviewer YjL7**
>
> Thank you for the detailed review and thoughtful feedback. Below we answer specific questions.
>
> >**How the method performs on different domain shifts. Does it perform comparably with Elastic Transform domain shift as it does with Fog?**
>
> - Yes, the proposed method consistently improves generalization under various shifts and corruptions. Please find the experimental results on different domains in the supplementary material (Table D.3-5).
>
> >**Why not compare results on the ImageNet-C dataset?**
>
> - As discussed in L312-313, a good estimate of moments often requires a handful of samples per class. As such, our current feature alignment strategy needs a batch size of over 1000 for ImageNet-C, which is beyond our hardware budget.
> - However, we would like to highlight that the proposed moment matching is the very first instance of our online feature alignment framework. Modifications on top of the current strategy are expected to scale better and can be explored in future work, e.g., decoupling the sample size from the batch size [1] or employing other forms of feature summarization [2].
> - We would also like to draw attention to the fact that scaling to ImageNet-C is a common challenge for recent test-time algorithms (e.g. SHOT). The prior methods (TENT, AdaptBN) shown most effective on ImageNet-C are often not competitive on other problems, e.g., VisDA-C, as reported in Table 2. The reason is that these methods are very restrictive about the freedom of adaptation and hardly generalize to large natural shifts. In contrast, our proposed method is more generic and flexible.
>
> >**It would be good to have a graph of classification error as test time training progresses similar to figure 2 from TTT Sun et al.**
>
> - Thanks for the suggestion. The table below summarizes the error rates of the TTT baseline and our proposed TTT++ as test-time training processes on CIFAR10 with the level-5 snow corruption. We will add detailed graphs in our updated version.
>
> 	| Number of Samples    |    500   |    1000    |    1500    |    2000   |   2500     |
> 	|:----------:|:--------:|:--------:|:--------:|:--------:|:--------:|
> 	| TTT-C 	 |   14.49   |   13.09   |   12.74   |   12.57   |   12.44   |
> 	| TTT++ (Ours) |   13.26   |   12.16   |   11.72   |  11.48   |   10.95   |
>
> >**Assumption that test feature distributions are aligned seems a bit strong.**
>
> - We would like to clarify that this assumption is made for theoretical analysis. In practice, the closer the test marginal feature distribution is to the training one, the better result is expected.
>
> >**Would it be possible to match all the moments of the distribution instead of only 2 moments?**
>
> - Yes, the proposed framework of offline summarization + online alignment is a general recipe and can be instantiated to incorporate high order moments.
> - In principle, the more moments are taken into account, the stronger empirical results are expected.
> - However, estimating higher-order moments often comes at an extra computational cost. Besides, the performance gain often gradually decreases as the order of moment increases, as discussed in [3,4].
>
> [1] Momentum Contrast for Unsupervised Visual Representation Learning, CVPR'20\
> [2] Prototypical Contrastive Learning of Unsupervised Representations, ICLR'21\
> [3] Central Moment Discrepancy (CMD) for Domain-Invariant Representation Learning, ICLR'17\
> [4] HoMM: Higher-order Moment Matching for Unsupervised Domain Adaptation, AAAI'20

---

> > ### Comment · Reviewer_YjL7 · 2021-08-30
> > **Update in the score**
> >
> > Authors have clarified some of my comments and for that reason I am increasing the score to "6: Marginally above the acceptance threshold". The reason I can not increase the score further is because - fundamentally the approach seems very similar to TTT where only major change is the change in the pretext task. As reviewer qJr7 also points this out - most of the performance gains is due to stronger and well established self-supervised contrastive objective.

---

> > > ### Author Response · Authors · 2021-08-31
> > > **Response to Reviewer YjL7's Additional Comments**
> > >
> > > Thanks for the comments! Indeed, our approach is heavily built upon TTT. Our main goal is to provide (i) an in-depth analysis and (ii) an improved version of it with (iii) more theoretical grounds.
> > >
> > > We appreciate the feedback on the performance gain. In fact, by incorporating the `decoupling` trick mentioned in our earlier response, the online feature alignment can bring performance gains as substantial as the strong SSL task, as summarized in our additional general response #2.

---

### Author Response · Authors · 2021-08-10
**General Response and Update**

We thank all reviewers for the thoughtful and constructive feedback. Below we address the common questions raised by reviewers.

**I. Clarification of Contributions**

As noted by Reviewers YjL7 & qJr7, we would like to reiterate that our main goal in this work is to provide **an in-depth analysis of the limitation and potential of the TTT framework**:

- We draw attention to the risk of feature misalignment, which is critical but largely overlooked in recent test-time algorithms.
- We shed light on the strong potential of the TTT framework by analyzing the nonlinear growth of test accuracy given improved SSL tasks.

These analyses inspire our proposed modification, which is simple yet effective, yielding state-of-the-art results on multiple robustness benchmarks.

**II. Motivating Example Update**

As suggested by Reviewer qJr7 & qJ8x, we will remove the illustration in Fig.1 and use the following simulation to build the rationale.

Our new motivating example is based on the two inter-twinning moons problem [1,2], where the main task is to predict the moon class of a given data point and the self-supervised task is to predict on which side of the hyperplane (i.e., linear separator between the two moons) the data point lies on, as shown below:
```
                      #
              #               #
        #                           #
    #                [x]                #                      [x]
[#]                      x                [#]                x
                             x                           x
                                   x               x
                                           x
```
*Note: The binary labels in the main task and the self-supervised task are identical for most data points, except the ones marked by [.]*

To illustrate the limitation of the vanilla TTT (Sec. 3), we simulate a large distributional shift through spatial translation and rotation of all data points. The results below demonstrate the importance of online feature alignment for test-time training.

|     Method    |     Accuracy    |
|:----------------:|:---------------:|
|       Train      |       100%      |
|        Test      |        50%      |
|        TTT       |        50%      |
|        TFA       |        50%      |
|  TTT++ (Ours)  |     87%     |

To illustrate how the distance between the main and SSL tasks affects the proposed TTT++ (Sec. 4), we further vary the separation distance between the two moons. The simulation result below is highly in line with our theoretical analysis.

| Separation Distance | TTT++ Accuracy |
|:---------------:|:------------:|
|       -0.1      |      87%      |
|       -0.2      |      84%      |
|       -0.3      |      67%      |
|       -0.4      |      58%      |

*Note: The binary labels in the main task and the self-supervised task are completely identical when the separation distance $\geq 0$*

[1] A PAC-Bayesian Approach for Domain Adaptation with Specialization to Linear Classifiers, ICML'13 \
[2] Domain-Adversarial Training of Neural Networks. JMLR'16

**III. Choice of Hyperparamter**

For both the proposed method and baselines, we performed a grid search on the key hyperparameters including the learning rate, batch size and weight coefficients using a validation set. The hyperparameters were kept constant for all types of image corruptions. Please refer to our submitted code for more details.

---

> ### Comment · Reviewer_qJr7 · 2021-08-13
> **Follow-up questions**
>
> Thanks for providing additional details! Two quick follow-up questions:
>  * "the nonlinear growth of test accuracy given improved SSL tasks": in which sense is the growth "nonlinear"?
>  * "grid search on the key hyperparameters including the learning rate, batch size and weight coefficients using a validation set": this refers to a labelled dataset, right? could the authors clarify how this fits into the setting of "unsupervised test time adaptation"? Generally, when doing hyperparameter tuning on labelled data, the _number of hyperparameters_ can be a confounding factor, that is: methods with more hyperparameters have an unfair advantage since they have more parameters that are tuned based on the (in principle) unavailable labelled data. And TTT+ has one more hyperparameter ($\lambda_z$) compared to TTT.

---

> > ### Author Response · Authors · 2021-08-14
> > **Follow-up Response**
> >
> > > **In which sense is the growth "nonlinear"?**
> >
> > * As noted in Eq.6 and Line 185, under certain assumptions, $\DeclareMathOperator{\E}{\mathbb{E}}
> > acc = \E_{y_s} \sum_{y_m} P(y_m | y_s) ^ 2$. \
> > The squared term implies the nonlinear growth of accuracy given improved SSL tasks.
> > * This can be more clearly seen in simple binary scenarios, e.g., the two moon example described above. \
> > $acc = 0.5 [ ( p_{m,+} | p_{s,+} ) ^ 2 + ( p_{m,-} | p_{s,+} ) ^ 2 ] + 0.5 [ ( p_{m,+} | p_{s,-} ) ^ 2 + ( p_{m,-} | p_{s,-} ) ^ 2 ] $ \
> > Let $p$ denote the probability of agreement between the main and SSL labels. \
> > $acc = 0.5 [ p ^ 2 + ( 1 - p ) ^ 2] + 0.5 [ (1 - p) ^ 2 + p ^ 2] = p ^ 2 + ( 1 - p ) ^ 2 = 2 (p - 0.5) ^ 2 + 0.5$ \
> > Without loss of generality, consider the case where the labels in the main and SSL tasks are positively correlated, i.e., $p \in (0.5, 1.0]$. The equation above suggests that improving the SSL task ($p = 0.5 \rightarrow p = 1.0$) leads to a quadratic growth of the test-time accuracy.
> >
> > > **Hyperparameters and confounding factors in unsupervised test time adaptation**
> >
> > * As noted in Lines 269, 323, 495, we follow the evaluation protocol in the previous work (TTT, SHOT, TENT): the hyperparameters are *not* tuned on each test domain. Instead, they are tuned on a different validation domain (cf. the `prepare_mix_corruption` function in `prepare_dataset.py`) and then kept fixed across all test domains (e.g., corruptions, severities). In other words, hyperparameters are not (confounding) variables at test time.
> > * It is true that our method introduces additional hyperparameters (weight parameters for feature alignment). However, since hyperparameters are tuned in the same way for all methods, as explained above, we do not think that additional hyperparameters make the comparison unfair. We'd be delighted to hear further questions and suggestions about evaluation protocols.

---

> > > ### Comment · Reviewer_qJr7 · 2021-08-17
> > > **Thanks for clarification**
> > >
> > > Thanks for clarifying these points. I recommend to add more details on the evaluation protocol to the main paper and not just referring to prior work since the protocol is an essential part of assessing the contribution of this work.

---

> > > > ### Author Response · Authors · 2021-08-17
> > > > **Thanks for the suggestions**
> > > >
> > > > Thanks for the suggestions! We will describe it in more detail in our updated version.

---

### Author Response · Authors · 2021-08-31
**General Response #2 amid Rolling Discussions**

We deeply appreciate all reviewers for valuable feedback, which has definitely helped improve our paper. \
In response to the comments raised amid rolling discussions, we’d like to briefly share the following analyses.

**I. Scalability of Feature Alignment**

As noted in our earlier responses to Reviewers YjL7 & qJr7, our proposed feature alignment strategy is expected to scale better to problems with a large number of classes, by `decoupling the sample size from the batch size for moment estimates`. We verify this by maintaining a dynamic queue of feature vectors progressively updated in a mini-batch manner. The table below summarizes the test errors obtained from the online feature alignment (TFA) on CIFAR100 with the level 5 fog corruption. The result confirms that, given a limited batch size, integrating a dynamic queue can enhance the scalability of our proposed feature alignment framework.

|                      |     |   |   w/o Queue  |      |  |  | |  w/ Queue |  |              |
|:-----------------:|:--:|:--------:|:-----:|:---------:|:--:|:--:|:--------:|:------:|:------:|:-------:|
| Sample Size |    |     64    |  128  |  256  |    |  | 64 x 2  | 64 x 4 | 64 x 8 | 64 x 16 |
|  Test Error    |     |   40.31   | 38.67 | 37.01 | | |   39.84  |  37.37 |  36.18 |  36.02  |

  *Note: Sample Size = [Batch Size] x [Number of Batches in the Queue]*

**II. Source of Performance Gains**

We agree with reviewers that, in some of our experiments, the online feature alignment was not the major source of the performance gain, which is largely due to the limited batch size we used under the single-gpu setup. To address this limitation, we integrate a large dynamic queue (64 * 16 or 128 * 16) for CIFAR-100, as described above, which allows us to obtain a more robust estimate of moments based on at least 10 (or 20) samples per class. As expected, our online feature alignment strategy (TFA) built with this simple modification leads to much lower classification errors than before, yielding highly competitive results on its own and significant performance gains on top of TTT-C.

|                  | brit | contr | defoc | elast |  fog | frost | gauss | glass | impul | jpeg | motn | pixel | shot | snow | zoom | Average |
|------------------|:----:|:-----:|:-----:|:-----:|:----:|:-----:|:-----:|:-----:|:-----:|:----:|:----:|:-----:|:----:|:----:|:----:|:-------:|
|       TTT-C      | **26.0** |  **27.7** | **31.0** |  42.1 | **36.2** |  38.6 |  42.2 |  49.6 |  46.7 | 36.5 | **33.1** |  34.1 | 41.0 | 39.3 | **28.8** |   36.9  |
|   TFA (64 * 16)  | 30.7 |  29.9 |  32.5 |  41.2 | 36.0 |  37.4 |  41.9 |  47.1 |  46.9 | 37.2 | 34.7 |  34.2 | 38.8 | 37.3 | 30.2 |   37.1  |
|  TFA (128 * 16)  | 28.8 |  29.4 |  **31.1** |  **40.4** | **35.9** | **36.8** | **40.3** | **44.7** | **42.6** | **35.6** | 33.6 |  **33.5** | **38.7** | **36.4** | 29.8 | **35.8** |
| TTT++ (128 * 16) | $25.4$ |  $26.6$ |  $29.1$ |  $39.2$ | $34.7$ |  $34.4$ |  $38.4$ |  $44.1$ | $41.3$ | $33.6$ | $32.9$ | $32.3$ | $37.7$ | $34.3$ | $27.8$ | $34.1$ |

  *Note: best result in $math$ font and major contributor in **bold***

We believe this result strengthens our finding that online feature alignment is a promising but overlooked component for test-time training, and complements the empirical evidence that we have shown in other settings:
* In the synthetic problem of large distributional shifts, TTT itself does not bring any improvements, unless combined with online feature alignment;
* On the robustness benchmark CIFAR10-C and VisDA, the online feature alignment contributes to ~10% error drops on top of the TTT-C. In addition, it plays a more crucial role in the low-data regime (e.g., 16 or 32 samples), as shown in our analysis suggested by Reviewer qJ8x.

We hope the analyses above bring more clarity with respect to our contributions. We will incorporate them into our updated version.

---

### Decision · Program_Chairs · 2021-09-27

**Decision:**

Accept (Poster)

**Comment:**

The work proposed a test time training method to address the distribution shift between training and testing using online feature alignment and contrastive loss based SSL. Reviewers overall see the paper well written and are content with the empirical evaluation. The rebuttal was able to convince some reviewers to raise their scores. I do agree with reviewer qJr7 that the contrastive loss based SSL task does not specifically address test time training, but rather is a generic approach to improve representation learning/generalization, which is well established. Focusing more of the discussion around the online feature alignment might shed more light on the new contribution of the work. As suggested by another reviewer, on-the-fly adaptation might be the more interesting setup to further distinguish the TTT setup from other setups facing distribution shift, such as domain adaptation.